# Enhanced specificity mutations perturb allosteric signaling in CRISPR-Cas9

Lukasz Nierzwicki[1†], Kyle W East[2†], Uriel N Morzan[3], Pablo R Arantes[1], Victor S Batista[4], George P Lisi[2*], Giulia Palermo[1*]

[1]Department of Bioengineering and Department of Chemistry, University of California, Riverside, Riverside, United States; [2]Department of Molecular Biology, Cell Biology and Biochemistry, Brown University, Providence, United States; [3]International Centre for Theoretical Physics, Trieste, Italy; [4]Department of Chemistry, Yale University, New Heaven, United States

**ABSTRACT** CRISPR-Cas9 (clustered regularly interspaced short palindromic repeat and associated Cas9 protein) is a molecular tool with transformative genome editing capabilities. At the molecular level, an intricate allosteric signaling is critical for DNA cleavage, but its role in the specificity enhancement of the Cas9 endonuclease is poorly understood. Here, multi-microsecond molecular dynamics is combined with solution NMR and graph theory-derived models to probe the allosteric role of key specificity-enhancing mutations. We show that mutations responsible for increasing the specificity of Cas9 alter the allosteric structure of the catalytic HNH domain, impacting the signal transmission from the DNA recognition region to the catalytic sites for cleavage. Specifically, the K855A mutation strongly disrupts the allosteric connectivity of the HNH domain, exerting the highest perturbation on the signaling transfer, while K810A and K848A result in more moderate effects on the allosteric communication. This differential perturbation of the allosteric signal correlates to the order of specificity enhancement (K855A > K848A ~ K810A) observed in biochemical studies, with the mutation achieving the highest specificity most strongly perturbing the signaling transfer. These findings suggest that alterations of the allosteric communication from DNA recognition to cleavage are critical to increasing the specificity of Cas9 and that allosteric hotspots can be targeted through mutational studies for improving the system's function.

**\*For correspondence:**
george_lisi@brown.edu (GPL);
giulia.palermo@ucr.edu (GP)

[†]These authors contributed equally to this work

**Competing interest:** The authors declare that no competing interests exist.

## Editor's evaluation

This paper presents an elegant and multidisciplinary study combining state-of-the-art NMR with computational modeling methods, to characterize the effects of mutations on the structure and allosteric communication within the CRISPR-Cas9 system. In revealing the link between the allosteric network in the protein and the increase in CRISPR-Cas9 specificity, this study carries important implications for the design of new gene editing tools.

## Introduction

CRISPR-Cas9 (clustered regularly interspaced short palindromic repeat and associated Cas9 protein) is a bacterial adaptive immune system with widely demonstrated and profound genome editing capabilities (*Doudna, 2020*). At the core of the CRISPR technology, the Cas9 endonuclease can be programmed with single-guide RNAs to site-specifically recognize and cleave any desired DNA sequence, enabling easy manipulation of the genome and playing a pivotal role in gene editing applications (*Jinek et al., 2012*). The RNA-programmable Cas9 generates double-stranded breaks in DNA by first binding complementary DNA sequences and then using two endonuclease domains, HNH and

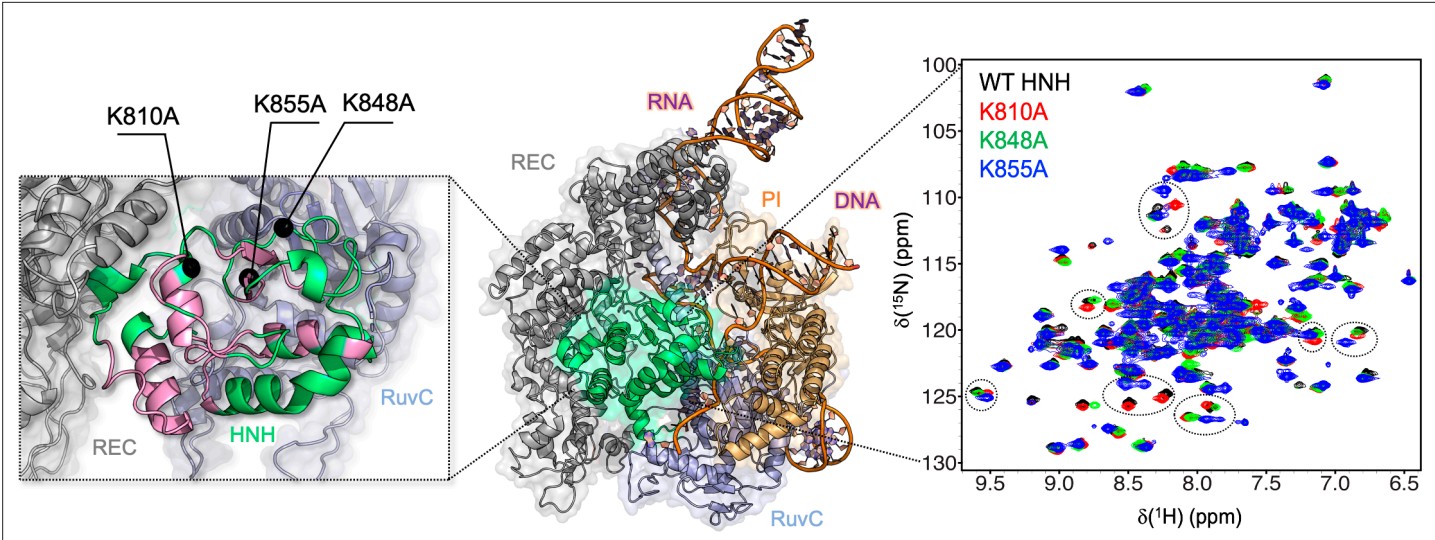

**Figure 1.** Architecture of the Cas9 endonuclease (center, PDB: 5F9R) (*Jiang et al., 2016*) highlighting its protein domains as follows: HNH (green), RuvC (light blue), PAM-interacting (PI, orange), and recognition lobe (REC, gray). A portion of the RNA:DNA hybrid behind HNH has been removed for clarity. Close-up view (left): the previously defined allosteric pathway spanning the HNH domain is shown (pink) and the locations of the three specificity enhancement K–*to*–A mutation sites are labeled (*East et al., 2020a*). Close-up view (right): $^1$H-$^{15}$N HSQC NMR spectra of K810A (red), K848A (green), and K855A (blue) overlaid with that of wild-type (WT) HNH (black). Selected areas of strong chemical shift perturbation are circled on the spectra (*Figure 1—figure supplements 1 and 2*).

The online version of this article includes the following figure supplement(s) for figure 1:

**Figure supplement 1.** Structure of the HNH domain.

**Figure supplement 2.** Circular dichroism of HNH mutants.

RuvC. Structures of *Streptococcus pyogenes* Cas9 (SpCas9) revealed that a large recognition lobe (REC) mediates the nucleic acid binding through three regions (REC1–3), while the spatially distinct HNH and RuvC nucleases act as molecular scissors on the two DNA strands (*Figure 1*; *Jiang and Doudna, 2017*).

Biophysical studies revealed that the molecular function of CRISPR-Cas9 is driven by an intricate allosteric communication, which is critical for transferring DNA binding information from the REC lobe to the catalytic sites for cleavage (*Sternberg et al., 2015* ; *Chen et al., 2017*; *Dagdas et al., 2017*; *Palermo et al., 2017*; *Palermo et al., 2018*). Biochemical and single-molecule experiments suggested that the catalytic HNH domain is the core of this allosteric relay (*Sternberg et al., 2015*; *Chen et al., 2017*). Indeed, the high flexibility of HNH can facilitate the signal transduction (*Jiang et al., 2016*; *Palermo et al., 2016*), exerting conformational control over double-stranded DNA cleavage (*Sternberg et al., 2015*). Solution NMR and all-atom molecular dynamics (MD) indicated a dynamic pathway of allosteric residues through HNH, depicting a mechanism for biological information transfer (*East et al., 2020a*). The contiguous dynamic network traverses the HNH domain, propagating the DNA binding signal from the REC region to the nucleases (HNH and RuvC, *Figure 1*, left panel) for concerted DNA cleavage (*Sternberg et al., 2015*). This provided a route for the allosteric transduction, and a mechanistic rationale for prior single-molecule and biochemical experiments (*Sternberg et al., 2015*; *Chen et al., 2017*), clarifying how HNH dynamics could transfer DNA binding signals from the REC region to the cleavage sites. This REC-HNH-RuvC allosteric communication is also critical for the system's specificity. Indeed, the binding of off-target DNA sequences at the REC lobe alters the dynamics of HNH and, in turn, affects the DNA cleavage capability of Cas9 (*Chen et al., 2017*; *Dagdas et al., 2017*; *Ricci et al., 2019*; *Mitchell et al., 2020*). To improve the system's specificity and reduce its off-target activity, extensive engineering of the Cas9 protein has been performed (*Chen et al., 2017*; *Kleinstiver et al., 2016*; *Slaymaker et al., 2016*; *Casini et al., 2018*). Three lysine-to-alanine (K-*to*-A) point mutations (i.e., K810A, K848A, and K855A) within the HNH domain have shown to be important for specificity enhancement (*Slaymaker et al., 2016*), but

their mode of action has remained unclear. This knowledge is of major importance, as it could help the mechanism-based design of improved Cas9 variants.

Here, we probe the structural and dynamic role of these mutations with respect to HNH allosteric signaling via molecular simulations, solution NMR, and network models derived from graph theory. This integrative approach holds the capability of defining allosteric motions with experimental accuracy through NMR (*Tzeng and Kalodimos, 2011*; *Lisi and Loria, 2016*; *Grutsch et al., 2016*; *Boulton and Melacini, 2016*) while also describing the network of communication with atomic level detail through computational methods (*Liu and Nussinov, 2016*; *Guo and Zhou, 2016*; *Dokholyan, 2016*; *East et al., 2020b*; *Vendruscolo, 2011*; *Wodak et al., 2019*; *Sethi et al., 2009*; *Bowerman and Wereszczynski, 2016*). We reveal that the three specificity-enhancing mutations alter the HNH allosteric structure, impacting the signal transmission from REC to RuvC. Moreover, while the K855A mutation strongly disrupts the REC to RuvC communication mediated by the HNH allosteric core, K810A and K848A result in more moderate effects. This difference in perturbing the allosteric signaling reflects the biochemical differences of the three mutants to increase Cas9 specificity (*Slaymaker et al., 2016*), suggesting that alterations of the allosteric pathway could be critical for the specificity enhancement. Taken together, our findings reveal that enhanced specificity mutations perturb the HNH allosterism, which in turn impacts Cas9 specificity. These findings represent a decisive step forward in understanding the role of allostery in the specificity of Cas9, and contributing engineering efforts toward improved genome editing.

## Results

Here, we harnessed the combination of MD simulations and solution NMR to describe the allosteric mechanism in CRISPR-Cas9 from its core (HNH) to the full complex. Solution NMR has been used to trace allosteric motions within a construct of the HNH domain that shows consistency with the structure of HNH from full-length CRISPR-Cas9 (*Figure 1—figure supplement 1*; *East et al., 2020a*). Indeed, due to the size of its polypeptide chain (i.e., 160 kDa), the Cas9 protein challenges traditional solution NMR, requiring optimized constructs of the individual domains to report on its structural and dynamical features (*East et al., 2020a*; *Nerli et al., 2021*). To address the allosteric mechanism, it is however critical to characterize the communication network within the full-length system. Toward this aim, we performed all-atom MD simulations of the full-length Cas9 protein in complex with RNA and DNA, employing model systems comprising >340,000 atoms. This enabled a 'bottom-up' approach, where the allosteric mechanism is evaluated from the individual HNH domain with experimental accuracy, to the full complex with atomic level detail through MD simulations.

### Structural perturbation of the HNH endonuclease

We used targeted mutagenesis to introduce the three previously identified specificity enhancement K-*to*-A mutations (K810A, K848A, and K855A) (*Slaymaker et al., 2016*) into the HNH construct (*East et al., 2020a*), and we employed solution NMR to determine the structural changes associated with these point mutations. First, changes to the local structure of HNH caused by each mutation were derived from chemical shift perturbations ($\Delta\delta$) in $^1$H-$^{15}$N HSQC NMR spectra (*Figure 1*, right panel). The overall structure of HNH is maintained in the mutants, as also confirmed by circular dichroism analysis revealing that all systems are >95% folded at the temperature of the experiments (25°C, *Figure 1—figure supplement 2*). The environmental perturbations were calculated using the method of Bax and coworkers (*Delaglio et al., 1995*). For each mutant, the composite $\Delta\delta$ $^1$H-$^{15}$N as well as the total $\Delta\delta$ are reported in *Figure 2*. Based on NMR measurements, it is apparent that each point mutation has a unique effect on the HNH domain. In detail, K855A displays severe exchange broadening (where the signal-to-noise has decreased by over 20-fold, *Figure 2A*, gray vertical bars) throughout the core of HNH (residues 842, 848, 849, 851, and 858, *Figure 2B*) and many significant $\Delta\delta$ on the RuvC-adjacent interface (residues 780, 813, 821–828, 838–841, 850, 853, 856–872, and 903–908). These regions are an integral part of the previously identified HNH allosteric pathway (*Figure 2B*; *East et al., 2020a*). K810A decreases the $\Delta\delta$ perturbation (residues 812–813, 834, 840–846, and 856), while K848A causes very modest chemical shift effects (residues 849 and 896). Most notably, the overall degree of $\Delta\delta$ perturbation decreases from K855A > K810A > K848A (*Figure 2C*, black bars). Intriguingly, this trend in the structural perturbation mirrors the location of these residues with

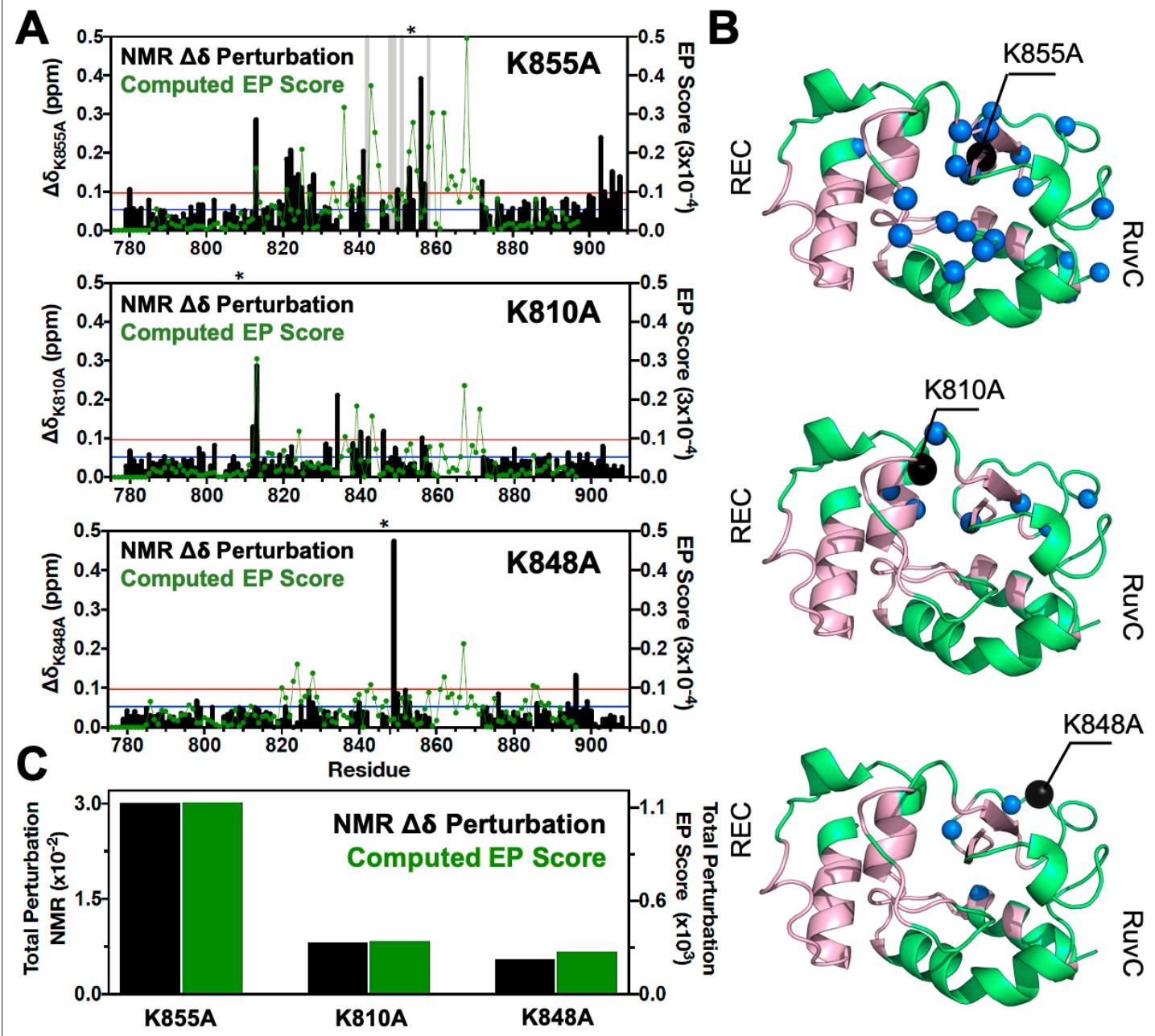

**Figure 2.** Environmental perturbations (EP) caused by the K855A, K810A, and K848A mutations in HNH. (**A**) The NMR composite chemical shifts

(black bars) determined by $\sqrt{\frac{1}{2}\left((\delta H)^2 + \frac{((\delta N)^2)}{25}\right)}$ are reported for each mutant. Blue lines are the 10% trimmed mean of all shifts and red lines

represent $1.5\sigma$ above the 10% trimmed mean. Sites of severe line broadening are represented by light gray bars. The site of mutation is indicated using an asterisk (*) above each plot. The EP scores computed for the isolated HNH domain from molecular dynamics (MD) simulations are shown using a green line. Blue and red lines from chemical shift analysis also represent the mean ($+1\sigma$) of the EP score data (right y-axis). (**B**) Residues with composite chemical shifts above the significance cutoff (blue spheres) are plotted onto the crystal structure of the HNH domain (green). The wild-type (WT) HNH allosteric pathway is also shown (pink). (**C**) Total chemical shift and EP (black and green bars, respectively) obtained as the sum of the NMR chemical shift perturbations and of the EP scores, respectively, for the isolated HNH domain (*Figure 2—figure supplement 1*).

The online version of this article includes the following figure supplement(s) for figure 2:

**Figure supplement 1.** Environmental perturbations (EP) caused by the K855A, K810A, and K848A mutations in HNH.

respect to the wild-type (WT) HNH allosteric pathway (*Figure 1*, left panel and 2B). Indeed, K855A, which is fully embedded in the allosteric pathway, shows the largest $\Delta\delta$ perturbation. K810A, on the periphery of the pathway, decreases the $\Delta\delta$ perturbation, while K848A, distant from the pathway, causes more moderate chemical shift effects. These observations suggest that K855A could have a more pronounced effect on the HNH allosteric structure, while K810A and K848A could exert a more moderate effect. This observation and its implications on HNH allosterism are thoroughly analyzed (vide infra).

## Perturbation of the chemical environment

We carried out all-atom MD simulations of CRISPR-Cas9 in its WT form (*Jiang et al., 2016*) and after introducing the K848A, K810A, and K855A mutations, obtaining replicas of μs trajectories and collecting a solid multi-μs statistical ensemble for the analysis of the allosteric mechanism (details are in the Materials and methods section) (*Palermo et al., 2017*; *East et al., 2020a*). Molecular simulations also considered the isolated HNH domain (*East et al., 2020a*) to assess the structure and dynamics of HNH. To determine the structural perturbation induced by the three K-*to*-A mutations from MD simulations, we introduced an environmental perturbation (EP) score, which determines the extent of the dynamic perturbation for a specific atom, given its local environment (details are in the Materials and methods section). We observe that the EP scores computed for the isolated HNH domain follow the experimental trend of the NMR $\Delta\delta$ in the HNH construct (*Figure 2A*). A qualitative agreement between the NMR and MD environmental perturbations is reasonable considering that the NMR $\Delta\delta$ are direct reporters of the local environment. Residues 860–870 remained unassigned in the NMR spectra, likely due to the remarkable flexibility of the region, which in turn results in high computed EP scores. It is also notable that the environmental perturbations computed for the HNH domain within full-length Cas9 are consistent with the EP scores of the HNH domain in its isolated form (*Figure 2*).

Overall, we observe that the total environmental perturbation arising from MD simulations follows the experimental trend of the NMR $\Delta\delta$, revealing that the structural perturbation is higher in K855A and decreases in K810A and K848A (*Figure 2C*). The qualitative agreement between the NMR $\Delta\delta$ and computed EP scores indicates that the structural ensemble obtained through all-atom MD for both the full-length Cas9 and the isolated HNH properly represents the local environment determined through the NMR $\Delta\delta$. The computed EP scores therefore support the idea that K855A could most strongly alter the allosteric structure of HNH, while K810A and K848A could progressively exert a more moderate impact.

## Alteration of the allosteric communication

Here, we combined Carr-Purcell-Meiboom-Gill (CPMG) relaxation dispersion NMR experiments with graph theory approaches to shed light on whether the three specificity enhancement mutations (i.e., K810A, K848A and K855A) could intervene in HNH-mediated allosteric communication. CPMG relaxation dispersion detects slow timescale (μs-to-ms) motions (*Loria et al., 1999*) that are indicative of allosteric signaling, as shown in a number of previous studies of allosteric enzymes (*Lisi and Loria, 2016*; *Oyen et al., 2017*). CPMG experiments on the three variants detected μs-ms motions in numerous residues comprising the WT pathway (e.g., G790, L791, I795, I841, F846, V856, S872, E873, I892, Q894, L900, *Figure 3—figure supplement 1*), with slight differences occurring in each of the variants (*Supplementary file 1*). The trend of curved dispersion profiles is consistent between WT HNH and the variants, suggesting the retention of dynamic residues comprising the allosteric pathway in the WT Cas9 (*East et al., 2020a*) with modest variations in the amplitude of the curves. Analysis of per-residue $R_{ex}$ values shows that on average, $\Delta R_{ex}$ between the three variants are <1.5. While regions of larger $\Delta R_{ex}$ appear when comparing the K-*to*-A variants to WT HNH, we note that the overall profiles are nearly identical for each mutant, with the largest $\Delta R_{ex}$ consistently occurring in residues 780–790 and surrounding residue 825, which are located in critical allosteric communities (vide infra). An analysis of exchange rates, $k_{ex}$, derived from CPMG experiments show that specificity-enhancing mutations skew the distribution of $k_{ex}$ toward slower regimes (*Figure 3—figure supplement 3*), while modestly reducing the average $k_{ex}$ for HNH overall. The $<k_{ex}>$ for WT HNH is ~1750 s$^{-1}$ (*East et al., 2020a*), while the mutants display similar $<k_{ex}>$ of 1507 s$^{-1}$ (K810A), 1640 s$^{-1}$ (K848A), and 1580 s$^{-1}$ (K855A). The average populations of ground and excited states are also similar for WT HNH and the K-*to*-A variants ($<p_a>$ 0.938 (WT), 0.931 (K810A), 0.937 (K848A), 0.940 (K855A) and $<\Delta\omega_N>$

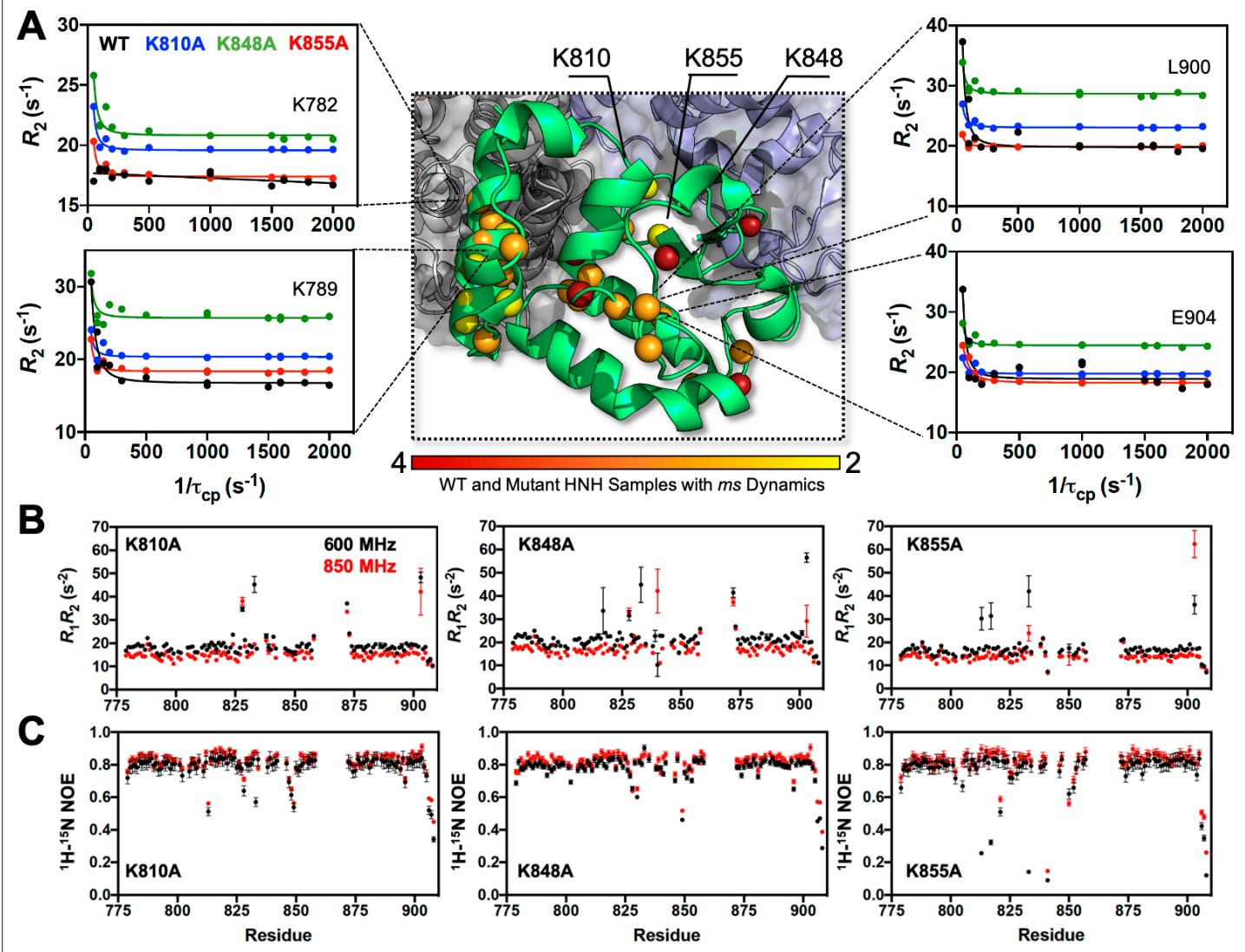

**Figure 3.** Dynamic properties of HNH and its mutants. (**A**) Structure of the HNH domain showing µs-ms timescale dynamics preserved following three K–to–A mutations (central panel). Spheres represent sites of Carr-Purcell-Meiboom-Gill (CPMG) relaxation dispersion that appear in the wild-type (WT) HNH and *at least one other* specificity enhancement variant. These sites preserve the dynamics upon mutations and are color-coded from red (highly preserved dynamics) to yellow (moderately, yet still preserved dynamics). Close-up views of representative CPMG relaxation dispersion profiles for WT (black), K810A (blue), K848A (green), and K855A (red) HNH are shown for various residues in this cluster. (**B**) Plots of the relaxation rate product, $R_1R_2$ for HNH mutants collected at 600 (black) and 850 (red) MHz. (**C**) Plots of the $^1$H-[$^{15}$N] heteronuclear nuclear overhouser effect (NOE) for HNH mutants collected at 600 (black) and 850 (red) MHz (**Figure 3—figure supplements 1–4**).

The online version of this article includes the following figure supplement(s) for figure 3:

**Figure supplement 1.** Dynamic properties of the HNH mutants.

**Figure supplement 2.** Per-residue $R_{ex}$ determined from Carr-Purcell-Meiboom-Gill (CPMG) relaxation dispersion NMR experiments for wild-type (WT) (black), K855A (red), K848A (green), and K810A (blue) HNH.

**Figure supplement 3.** Distribution of $k_{ex}$ values determined from Carr-Purcell-Meiboom-Gill (CPMG) relaxation dispersion for wild-type (WT) HNH and specificity-enhancing mutants.

**Figure supplement 4.** Order parameters ($S^2$) derived from model-free analysis of $T_1$, $T_2$, and $^1$H-[$^{15}$N] nuclear overhouser effect (NOE) experiments.

368 Hz (WT), 347 Hz (K810A), 275 Hz (K848A), and 326 Hz (K855A)). Likewise, faster (ps-ns) timescale motions are only locally altered by the mutations (**Figure 3B and C**), with good agreement between order parameters derived from WT HNH and all variants ($\Delta S^2 \leq 0.1$, **Figure 3—figure supplement 4**). There is also agreement between $S^2$ determined experimentally and computationally, consistent with

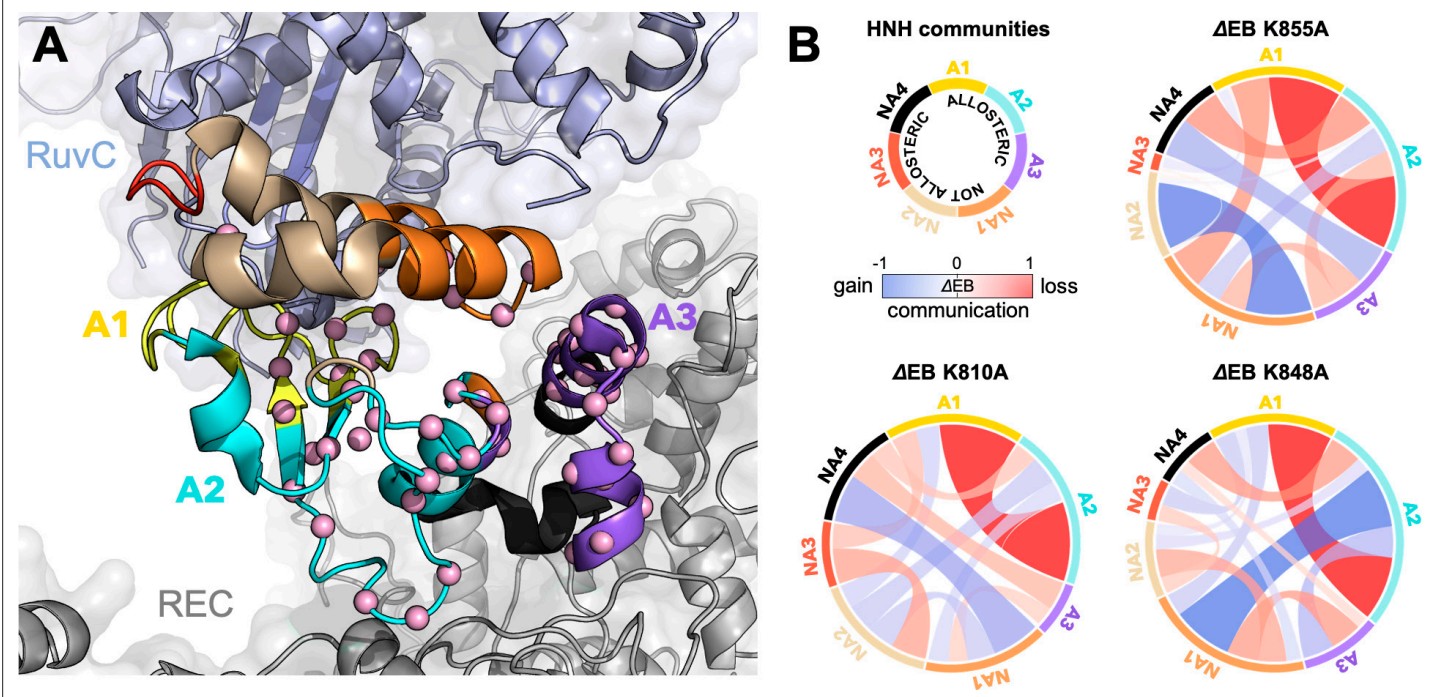

**Figure 4.** Alterations of the allosteric signaling in full-length CRISPR-Cas9 (clustered regularly interspaced short palindromic repeat and associated Cas9 protein) systems. (**A**) Close-up view of the HNH domain within the wild-type (WT) full-length Cas9, showing seven communities of synchronized dynamics, indicated using different colors. Three communities are allosteric (A1 yellow, A2 cyan, and A3 purple), since they hold most of the residues that compose the allosteric pathway (shown as spheres). The non-allosteric communities (NA1 orange, NA2 tan, NA3 red, NA4 black) include only a few allosteric residues. (**B**) Circular networks reporting the mutation-induced edge betweenness change (ΔEB), a measure of communication gain or loss between couples of communities upon mutation. For each of the K855A, K810A, and K848A mutants, the HNH communities are displayed in a circle and are connected by links with thickness proportional to ΔEB (border sizes also correspond to the ΔEB of the specific community). Negative ΔEB (red) represents loss of communication, positive ΔEB (blue) indicates communication gain upon mutation (**Figure 4—figure supplement 1**).

The online version of this article includes the following figure supplement(s) for figure 4:

**Figure supplement 1.** Circular graphs reporting the mutation-induced edge betweenness change (ΔEB) for the K855A, K810A, and K848A mutants in the HNH domain in the full-length Cas9 (**A**) and in its isolated form (**B**).

$^1$H-[$^{15}$N] NOEs that show depressed values sporadically between residues 800 and 825, surrounding residue 850, and at the C-terminus.

To describe the allosteric communication pathway, information theory was applied to the analysis of our µs-length simulations of the Cas9 variants. We computed the dynamic pathways composed of residue-to-residue steps that optimize the momentum transport (and thereby maximize the correlations) from REC to RuvC, as for the WT system (details are reported in the Materials and methods section) (**East et al., 2020a**). We found that the residues composing the dynamic pathways of the three variants differ very little from the WT, and are consistent between full-length CRISPR-Cas9 and the HNH construct ( **Appendix 1—figures 1–4**). This indicates that dynamic allosteric signaling is preserved in the Cas9 mutants, in agreement with CPMG relaxation experiments. In this respect, the pathways that maximize the dynamic transmission between RuvC and REC2 through HNH agree well with the µs-ms motions identified in the HNH construct via CPMG relaxation dispersion (**Figure 3—figure supplement 4**).

To further understand the effect of the K810A, K848A, and K855A mutations on the allosteric mechanism, we employed community network analysis (**Sethi et al., 2009**) to identify the groups of residues that comprise cohesive structural units with synchronized dynamics within HNH. These 'communities' of highly correlated residues establish a dynamic 'crosstalk' with each other, the strength of which can be quantified using the 'edge betweenness' (EB) measure (details are reported in the Materials and methods section). This analysis was performed on the full-length CRISPR-Cas9 systems (reported here), and for comparison, on the isolated HNH domain (reported in Appendix 1). First,

community network analysis performed on the WT HNH identified seven communities (*Figure 4A* and *Appendix 1—figure 5*). Three communities hold most of the residues that display slow timescale dynamics from CPMG relaxation dispersion, and that compose the allosteric pathway in the WT Cas9 (*East et al., 2020a*). These 'allosteric' communities (A1, yellow; A2, cyan; and A3, purple) are part of the allosteric route communicating the RuvC and REC2 interfaces (at the A1 and A3 communities, respectively). The 'non-allosteric' communities (NA1, orange; NA2, tan; NA3, red; and NA4, black) include only few allosteric residues. To understand how each mutation affects the inter-community crosstalk and the allosteric network, we analyzed the EB of the WT HNH and its mutants and we computed the mutation-induced EB change (ΔEB, details are in the Materials and methods section and in *Appendix 1—figures 6–8*), allowing us to quantify the perturbation in the communication. We introduced circular networks of the ΔEB (*Figure 4B*), where the communities are displayed in a circle and connected using links with thickness proportional to the ΔEB. Negative values of ΔEB (red) represent loss of communication, as opposed to positive values (blue), which indicate a communication gain upon mutation. For all mutants, we observe a dramatic loss of communication between the A1 and A2 allosteric communities, which are central to the allosteric pathway (*Figure 4B*). This loss of communication at the core of the HNH allosteric structure indicates that the substitutions that enhance the specificity of Cas9 disrupt the allosteric crosstalk between RuvC and REC2. This evidence links the enhancement of specificity to the disruption of the allosteric pathway, pinpointing that an increase in specificity induced by the K-*to*-A mutations is associated with alteration of the HNH allosteric signaling. This is consistent with $\Delta R_{ex}$ plots from NMR data (*Figure 3—figure supplement 2*) that show a loss of flexibility (negative $\Delta R_{ex}$) in critical regions of the K-*to*-A variants, particularly A1 and A3.

The three K-*to*-A mutations also display important differences. Indeed, K855A mainly disrupts the crosstalk between the allosteric communities, while the non-allosteric sites gain communication. The two other mutants, K810A and K848A, display a progressive gain in communication between both allosteric and non-allosteric sites. In these mutants, the allosteric A1 and A2 communities gain communication with the non-allosteric NA1 and NA2 regions. This effect reduces the negative impact of these mutations on the allosteric core of HNH. Analyses of the isolated HNH domain are consistent with these findings (*Figure 4—figure supplement 1*), confirming that K855A mainly disrupts the communication between sites that are highly involved in the allosteric pathway, while K810A and K848A also display gain in communication between allosteric and non-allosteric sites.

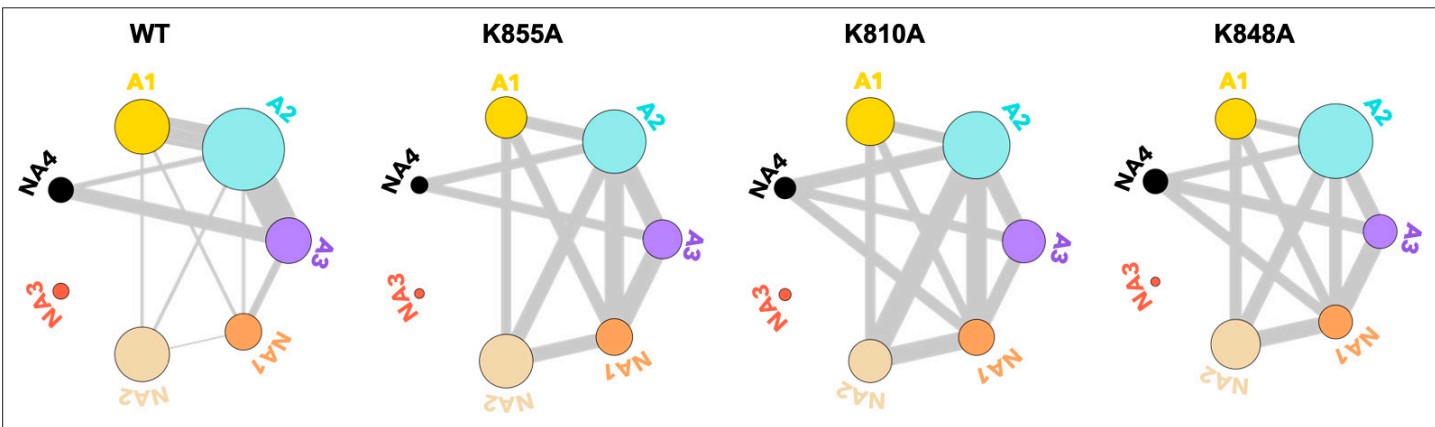

**Figure 5.** Residues exchanging between dynamic communities of the wild-type (WT) HNH domain and its mutants in full-length CRISPR-Cas9 (clustered regularly interspaced short palindromic repeat and associated Cas9 protein). The HNH communities identified through computational analysis are used as a reference, while the dynamic exchange among them is based on the number of residues displaying Carr-Purcell-Meiboom-Gill (CPMG) relaxation dispersion. The allosteric communities (A1–A3) hold most of the residues that compose the WT allosteric pathway, while the non-allosteric communities (NA1–NA4) include only a few allosteric residues. Bond thickness reflects a normalized measure of residues with μs-ms motions exchanging between communities, where thicker lines indicate that a greater number of CPMG-detectable dynamics (details are in the Materials and methods section) (*Figure 5—figure supplement 1*).

The online version of this article includes the following figure supplement(s) for figure 5:

**Figure supplement 1.** Residues exchanging between dynamic communities of the wild-type (WT) HNH domain and its mutants in the full-length CRISPR-Cas9 (clustered regularly interspaced short palindromic repeat and associated Cas9 protein) system (**A**) and in the isolated HNH domain (**B**).

To further evaluate how the dynamic exchange between the HNH communities is altered by the K-*to*-A mutations, we analyzed our NMR CPMG relaxation dispersion data in the context of the community networks (details are reported in the Materials and methods section). In this analysis, the computationally derived HNH communities are used as a reference, while the dynamic exchange among them is derived from CPMG relaxation dispersion. In *Figure 5*, we show the communities of HNH in the full-length CRISPR-Cas9 system and its mutants, where the thickness of the connecting bonds represents dynamic exchange arising from residues undergoing CPMG relaxation dispersion.

From this analysis of the NMR data, we detect a decrease in the dynamic exchange between the allosteric communities upon mutation (represented by thinner bonds compared to the WT), while an increase is observed with the non-allosteric communities. This trend is consistent with the analysis of CPMG data with respect to the communities of the isolated HNH domain (*Figure 5—figure supplement 1*). Overall, the reduction of dynamic exchange between allosteric communities and greater involvement of non-allosteric communities in the mutants is consistent with the mutation-induced EB determined from computational analysis (*Figure 4*).

In summary, our results show that three K-*to*-A mutations alter the HNH allosteric structure, with K855A exerting the most pronounced perturbation of the signal transfer by several metrics. This links the specificity enhancement of Cas9 to alterations of its allosteric crosstalk. As we discuss below, such alterations reflect the differential capability of the three point mutants to enhance Cas9 specificity (*Slaymaker et al., 2016*), offering important insight into the mechanistic basis of specificity enhancement, which is poorly understood and difficult to address.

## Discussion

Allostery is a fundamental property of the CRISPR-Cas9 gene editing tool (*Nierzwicki et al., 2021*; *Zuo and Liu, 2020*). In this system, the allosteric relay is critical for transferring DNA binding information from the recognition (REC) lobe to the nuclease domains for cleavage and specificity (*Sternberg et al., 2015*; *Chen et al., 2017*; *Dagdas et al., 2017*; *Palermo et al., 2017*; *Palermo et al., 2018*). Biochemical (*Sternberg et al., 2015*), structural (*Jiang et al., 2016*), and biophysical (*Chen et al., 2017*; *Dagdas et al., 2017*; *Palermo et al., 2017*; *Palermo et al., 2018*) approaches have shown that the HNH domain is the crux of this allosteric regulation, possessing a striking flexibility that facilitates the signal transduction (*Jiang et al., 2016*; *Palermo et al., 2016*) and controls DNA cleavage (*Sternberg et al., 2015*).

Here, we combined solution NMR, MD, and network theory to elucidate the allosteric role of three critical point mutations in HNH – K810A, K848A, and K855A – that increase the specificity of Cas9 and reduce its off-target activity (*Slaymaker et al., 2016*). We first analyzed the possible structural perturbations induced by the presence of these point mutations. The NMR $\Delta\delta$ revealed that K855A induces the most significant structural perturbations, while K810A and K848A progressively reveal weakened perturbations (*Figure 2*). This was consistent with the computational assessment of the chemical environment, reporting a similar trend and a decreased EP score from K855A > K810A > K848A (*Figure 2C*) and conveying that the three mutations differentially alter the HNH structure.

To characterize the allosteric signaling, we combined CPMG relaxation dispersion experiments with computational analyses based on graph theory that are suited for the detection of allosteric effects (*Liu and Nussinov, 2016*; *Guo and Zhou, 2016*; *Dokholyan, 2016*; *East et al., 2020b*; *Vendruscolo, 2011*; *Wodak et al., 2019*; *Sethi et al., 2009*; *Bowerman and Wereszczynski, 2016*). We found that the three mutants retain the overall dynamic pathway responsible for information transfer, indicating that the allosteric signaling is preserved (*Figure 3* and *Appendix 1—figure 4*). We then employed community network analysis (*Sethi et al., 2009*) to identify the communication structure and how it is rewired by the specificity-enhancing mutations. In-depth analysis of the community network (*Figure 4*) revealed that the K-*to*-A mutations disrupt the main communication channel between RuvC and REC, as evidenced by a decrease in the mutation-induced edge betweennesses difference (ΔEB) between the allosteric communities A1–A3 that are central to the pathway (*Figure 4B*). This is also evident in the integrated analysis of NMR relaxation data in context of the communities (*Figure 5*), showing a decrease in the dynamic exchange between the allosteric communities upon mutation. This indicates that an increase in specificity is associated with alterations of the HNH allosteric structure. Interestingly, computational and experimental data indicate that the K-*to*-A mutations consistently disrupt the crosstalk between allosteric sites A1 and A2 (*Figure 4*), suggesting that these communities could be

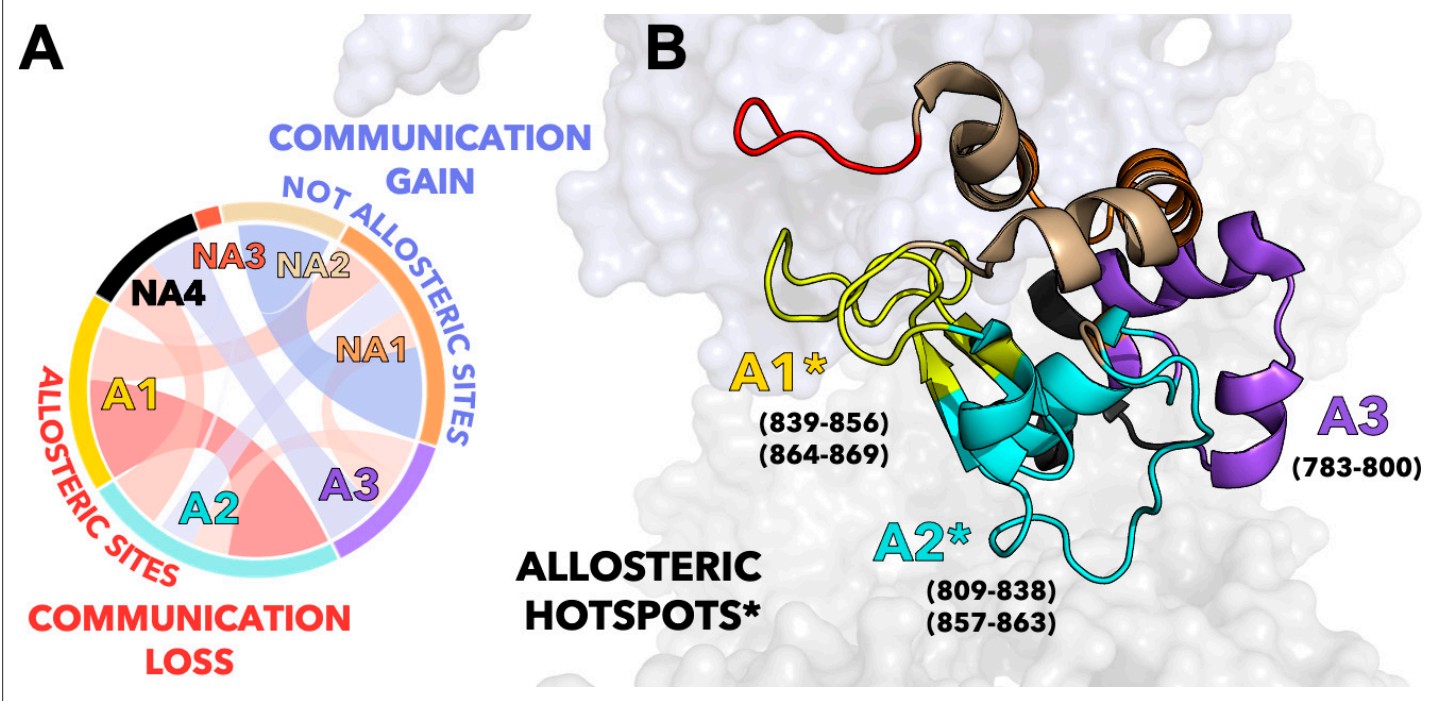

**Figure 6.** Critical allosteric hotspots of signal transmission through the HNH domain of CRISPR-Cas9 (clustered regularly interspaced short palindromic repeat and associated Cas9 protein). (**A**) Circular graph showing that allosteric sites of HNH (**A1–A3**) display a loss in communication (red bond) upon alanine mutation of K855, K810, and K848, while the non-allosteric sites (NA1–NA4) gain communication (blue bond). The three K-to-A mutations mainly disrupt the communication between A1 and A2, suggesting that these communities could be critical allosteric hotspots for the signal transmission. (**B**) The A1–A2 allosteric hotspots are shown on the 3D structure of HNH in CRISPR-Cas9 and indicated using an asterisk (*). Residues comprising the allosteric communities A1–A3 are reported.

critical allosteric hotspots for the signal transmission (*Figure 6*). Building on this observation, future mutational studies of residues comprising the A1–A2 communities (i.e., residues 839–856, 864–869, 809–828, and 857–863) could impact the allosteric communication and, in turn, modulate the function and specificity of the system.

In-depth analysis of the computational data also reveals that K855A more strongly disrupts the communication between allosteric sites, compared to K848A and K810A that display a lesser impact (*Figure 4B*). This is consistent with the structural perturbations observed through the NMR $\Delta\delta$ and the computed EP scores (*Figure 2*), showing that the dynamic perturbation induced by K855A mutant is more pronounced than that generated by K810A and K848A variants. Interestingly, among the three K-to-A mutations, K855A has shown to achieve the highest specificity enhancement as a single point mutation (*Slaymaker et al., 2016*). Indeed, the specificity enhancement of the three single mutants toward the off-target validating VEGFA gene follows the K855A > K848A ~ K810A order. In the same study, K810A and K848A required further optimization to achieve maximal specificity. Hence, the biophysical aspects of allosteric signal perturbation mirror the biochemistry of specificity enhancement in these three single point mutants.

On the basis of these observations, K855A, which strongly impacts allosteric communication, could most prominently leverage the HNH allosterism to improve the system's specificity. The HNH allosteric signal is indeed critical for transferring DNA binding information from the REC region to the catalytic sites of HNH and RuvC, and is a cornerstone of CRISPR-Cas9 specificity. Single-molecule and kinetic experiments (*Chen et al., 2017*; *Dagdas et al., 2017*), as well as computational analysis (*Palermo et al., 2018*; *Ricci et al., 2019*; *Mitchell et al., 2020*), have shown that the binding of off-target DNA sequences at REC affects the dynamics of HNH and its allosteric activation of DNA cleavage. Our investigations show that three specificity-enhancing mutations disrupt this signaling mechanism. Considering that the magnitude of change in allosteric communication correlates to the biochemical trends in specificity enhancement (K855A > K848A ~ K810A) (*East et al., 2020a*), the structural and dynamic perturbations caused by K-to-A mutants can be related to the enzyme's increased specificity.

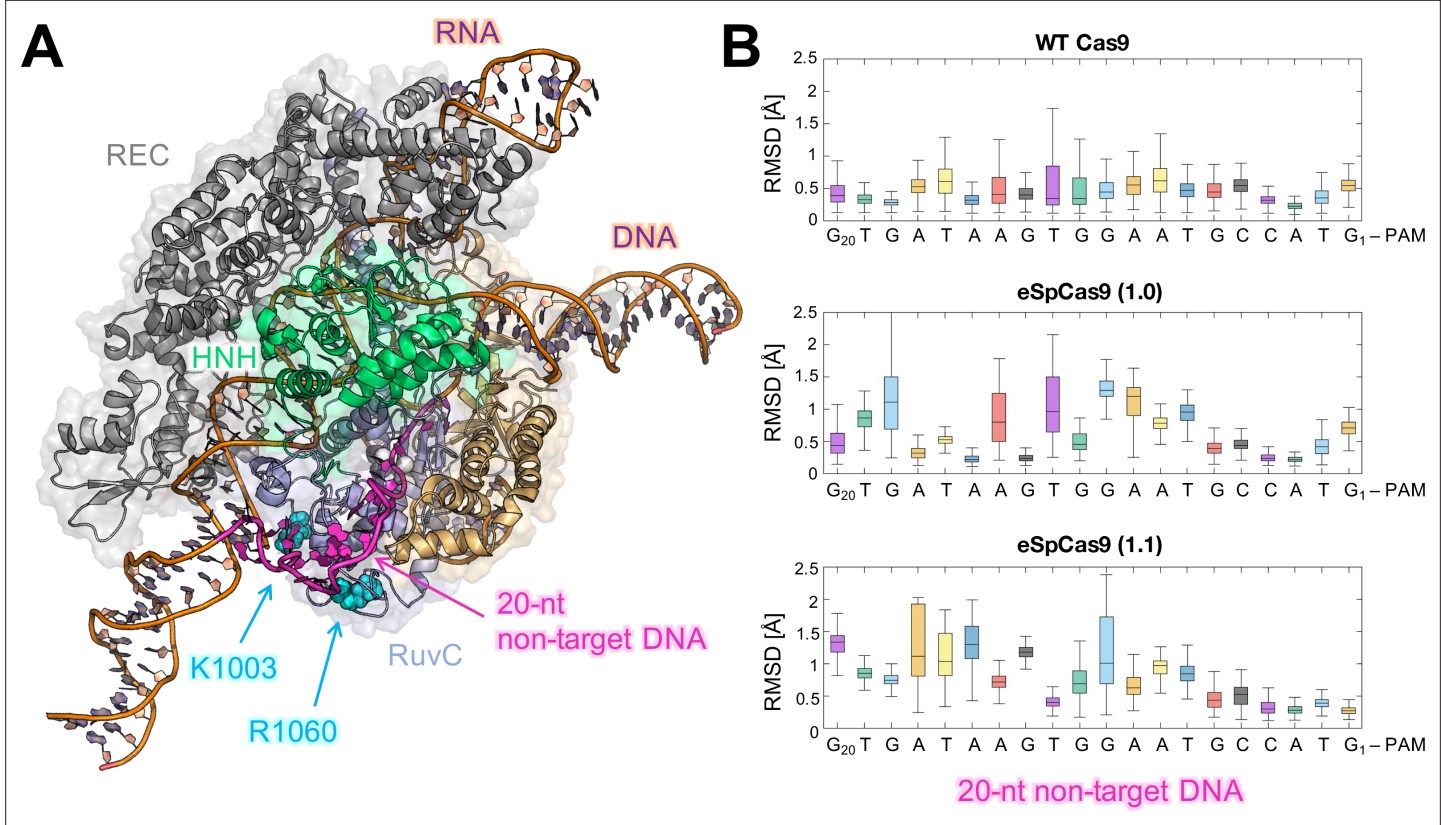

**Figure 7.** Flexibility of the DNA non-target strand. (A) Overview of an enlarged model system of the ild-type (WT) CRISPR-Cas9, including a longer DNA non-target strand reaching the K1003 and R1060 residues (PDB: 5Y36) (*Jiang and Doudna, 2017*). Cas9 is color-coded as in *Figure 1*. The nucleic acids are shown as ribbons, highlighting in magenta the 20 nucleotides (20-nt) segment of the DNA non-target strand that locates within the RuvC domain. (B) For each nucleotide of the 20-nt segment, the root mean square deviations (RMSD) of the heavy atoms with respect to the initial positions have been computed along the simulations of the WT Cas9 (top), SpCas9 1.0 (center), and SpCas9 1.1 (bottom) systems and are plotted using box plots. All analyses performed over the last ~1 μs of MD simulations reveal a remarkable increase in the flexibiliy of the DNA non-target strand in the SpCas9 1.0 (center), and SpCas9 1.1 variants.

The K810A and K848A mutants, which affect the HNH structure and dynamics in a more moderate way, could exert a lower allosteric effect on the specificity enhancement. As noted above, these mutants require additional substitutions to achieve maximal specificity (*Slaymaker et al., 2016*). The additional K1003A and R1060A mutations localized away from HNH (*Figure 7*) and might exploit different mechanisms to improve Cas9 specificity, such as altering the interactions with the distal bases of the DNA non-target strand (*Slaymaker et al., 2016*). To understand this further, we performed additional μs-length MD of the triple mutants K810A-K1003A-R1060A (viz., eSpCas9 1.0) and K848A-K1003A-R1060A (viz., eSpCas9 1.1), as well as of the WT Cas9 in an enlarged model system including a longer DNA non-target strand (*Figure 7* and S2. Supplementary Results). The K1003A and R1060A mutations induce a remarkable flexibility of the distal DNA bases, compared to the WT Cas9 (*Figure 7*), consistent with the hypothesis that the specificity enhancement of these mutants could also arise from the neutralization of positive charges interacting at the level of the DNA non-target strand (*Slaymaker et al., 2016*). This could indeed favor the re-hybridization of DNA in the presence of off-target sequences, thereby limiting off-target cleavage (*Singh et al., 2016* ; *Singh et al., 2018*).

Taken together, our observations suggest that K855A could increase the specificity by mainly leveraging the HNH allosterism, while K810A and K848A could combine more moderate allosteric effects with the weakening of the interactions at the DNA non-target strand. This combination of allosteric and electrostatic effects could be critical for the triple mutants, which were optimized for both specificity and activity, with eSpCas9 1.1 being widely used in vitro (*Slaymaker et al., 2016*). Our investigations show that, in addition to electrostatic effects, the K-*to*-A mutants alter HNH dynamics, impacting the allosteric communication between REC and RuvC (*Sternberg et al., 2015*; *Chen et al., 2017*;

*Dagdas et al., 2017*; *Palermo et al., 2017*; *Palermo et al., 2018*). This altered communication has a profound influence on Cas9 activation, representing an important source for the observed specificity enhancement. We therefore suggest that the allosteric crosstalk can be targeted for improving the system's specificity, as reported for other allosteric proteins (*Liu and Nussinov, 2016*). In this respect, the critical hotspots identified here (*Figure 6*) offer novel insight for mutational studies of CRISPR-Cas9, aimed at further controlling its function.

Finally, it is worth noting that the combination of solution NMR and molecular simulations enabled us to translate the allosteric signaling from the individual HNH domain to full-length Cas9 with atomic level detail. This 'bottom-up' approach exploits the capability of solution NMR to identify allosteric motions within optimized constructs of the multi-domain Cas9 protein (*East et al., 2020a*; *Nerli et al., 2021*), while all-atom MD simulations are used to characterize the communication network within the full-length Cas9. Future studies in our laboratories will leverage this approach to fully characterize the allosteric transmission across the multiple domains of Cas9 and its variants, gaining thorough insight into the system's function and specificity.

## Conclusions

Here, molecular simulations in combination with solution NMR and graph theory revealed that three lysine-to-alanine point mutations, which substantially increase the system's specificity (*Slaymaker et al., 2016*), alter the allosteric mechanism of information transfer in the CRISPR-Cas9 HNH endonuclease, impacting the signal transmission from the DNA recognition region to the catalytic sites. Among the three K855A, K810A, and K848A specificity-enhancing mutations, K855A strongly disrupts the HNH domain allosteric structure, exerting the highest perturbation on the signaling transfer, while K810A and K848A result in more moderate effects on the allosteric intercommunication. This differential perturbation of the allosteric signaling reflects the different capabilities of the single mutants to increase Cas9 specificity biochemically, with the mutation achieving the highest specificity most strongly perturbing the signaling transfer. Considering that the information transfer from DNA recognition to cleavage is critical for the system's function (*Sternberg et al., 2015*; *Chen et al., 2017*) and its specificity against off-target effects (*Chen et al., 2017*; *Dagdas et al., 2017*), the structural and dynamic perturbations caused by the three K-*to*-A mutants can be related to the enzyme's increased specificity. Taken together, these findings are a step forward in the molecular level understanding of the CRISPR-Cas9 mechanism, and open the door for harnessing the allosteric signaling toward the improved system's specificity.

# Materials and methods
## Protein expression and purification

The three specificity enhancement mutations (K810A, K848A, and K855A) of the SpCas9 protein were introduced into a previously reported construct of the HNH domain (residues 775–908) that has shown consistency with the structure of HNH within the full-length CRISPR-Cas9 system (*East et al., 2020a*; *Slaymaker et al., 2016*). N-labeled NMR samples were expressed in Rosetta(DE3) cells in M9 minimal containing MEM vitamins, $MgSO_4$, and $CaCl_2$. Cells were induced with 0.5 mM IPTG after reaching an OD600 of 0.8–0.9 and grown for 16–18 hr at 20°C post-induction. Cells were harvested by centrifugation and resuspended in 20 mM HEPES, 500 mM KCl, and 5 mM imidazole at pH 8.0. Cells were then lysed by ultrasonication and purified on Ni-NTA column with an elution buffer of 20 mM HEPES, 250 mM KCl, and 220 mM imidazole at pH 7.4. The N-terminal His-tag was removed by TEV protease. NMR samples were dialyzed into a buffer containing 20 mM HEPES, 80 mM KCl, 1 mM DTT, 1 mM EDTA, and 7.5% (v/v) $D_2O$ at pH 7.4.

## NMR spectroscopy

NMR spin relaxation experiments were carried out at 600 and 850 MHz on Bruker Avance NEO and Avance III HD spectrometers, respectively. All NMR spectra were processed with NMRPipe (*Delaglio et al., 1995*) and analyzed in NMRFAM-SPARKY (*Lee et al., 2015*). CPMG (*Loria et al., 1999*) experiments were adapted from the report of Palmer and coworkers with a constant relaxation period of 40 ms and $v_{CPMG}$ values of 25, 50 × 2, 100, 150, 200, 400, 500 × 2, 600, 800 × 2, 900, 1000 Hz. Relaxation dispersion curves were generated and exchange parameters were obtained from fits of the individual

data carried out with RELAX (*Bieri et al., 2011*) using the R2eff, NoRex, Tollinger (TSMFK01), and Carver-Richards (CR72 and CR72-Full) models. Longitudinal and transverse relaxation rates were measured with relaxation times of 20 × 2, 60 × 2, 80, 200 × 2, 400, 800, 1000, and 1200 ms for $T_1$ and 8.48, 16.96 × 2, 33.92, 67.84, 84.8 × 2, 101.76 × 2, 118.72, 135.68 for $T_2$, where $x2$ represents duplicate relaxation times (*Luz and Meiboom, 1963*). Peak intensities were quantified in SPARKY and the resulting exponential decays were fit in Mathematica. Steady-state $^1$H-[$^{15}$N] nuclear overhouser effect (NOE) were measured with a 6 s relaxation delay followed by a 3 s saturation (delay) for the saturated (unsaturated) experiments. All relaxation experiments were carried out in a temperature-compensated, interleaved manner. Model-free analysis using the Lipari-Szabo formalism (*Lipari and Szabo, 2002*) was carried out on dual-field NMR data in RELAX with fully automated protocols.

## MD simulations

MD simulations were performed on the full-length CRISPR-Cas9 system and on the isolated HNH domain. The X-ray structures of the full-length Sp CRISPR-Cas9 (5F9R.pdb, at 3.40 Å) (*Jiang et al., 2016*) and of the HNH construct (6O56.pdb, 1.90 Å resolution) (*East et al., 2020a*) were used as models. Both systems were considered as WT and with three single-point mutations K810A, K848A, and K855A (*Slaymaker et al., 2016*), resulting in eight simulation systems. All systems were solvated reaching periodic boxes of ~34,000 (isolated HNH) and ~340,000 (full-length Cas9) total atoms. A new AMBER ff99SBnmr2 force field (*Yu et al., 2020*), which improves the consistency of the backbone conformational ensemble with NMR experiments, was used for the protein. Nucleic acids were described, including the ff99bsc0+ $\chi$ OL3 corrections for DNA (*Pérez et al., 2007*) and RNA (*Zgarbová et al., 2011*; *Banáš et al., 2010*). The TIP3P model was used for water (*Jorgensen et al., 1983*). Simulations were performed in the NPT ensemble with temperature held at 310 K using the Bussi thermostat (*Bussi et al., 2007*). The pressure was held at 1 bar with the Parrinello-Rahman barostat (*Parrinello and Rahman, 1981*). A time step of 2 fs was applied (details are reported in the SI). An ~1.2 µs-long trajectory was collected in three replicas for the WT CRISPR-Cas9 system and for each of the K845A, K855A, and H855A variants. The isolated HNH domain was also simulated in three replicas of ~1.2 µs each as WT and introducing the three K-*to*-A point mutations. This resulted in ~3.6 µs of MD for each simulated system and a total of ~14.4 µs of MD for the full-length CRISPR-Cas9 and also for the isolated HNH domain. This simulation length (in three replicates) was motivated by our previous work (*Palermo et al., 2017*; *East et al., 2020a*), showing that it provides a solid statistical ensemble for the purpose of the analysis of the allosteric mechanism (described below). Analysis of the results was performed after discarding the first ~200 ns of MD, to enable proper equilibration and a fair comparison. Data are reported for the overall ensemble in the main text and for the separated replicas in Appendix 1. Three additional systems, including a longer DNA non-target strand, were also built to simulate the triple mutants K810A-K1003A-R1060A (eSpCas9 1.0) and K848A-K1003A-R1060A (eSpCas9 1.1) (*Slaymaker et al., 2016*) and the WT Cas9 (details are reported in Appendix 1). These systems were based on the cryo-EM structure EMD-8236 (5Y36.pdb, at 5.20 Å resolution) (*Huai et al., 2017*), since it provides structural information of the terminal bases of the DNA non-target strand. These solvated systems comprised ~412,000 atoms and were also simulated for ~1.2 µs each. All simulations were performed using Gromacs (v. 2018.3) (*Van Der Spoel et al., 2005*).

## EP score

To determine the structural perturbation induced by the three K-*to*-A mutations from MD simulations, we introduced a new EP score measure. The EP score is a measure of the mutation-induced dynamic perturbation experienced by each residue in HNH given its local environment. The EP score analysis begins with the definition of a threshold radius $r_t$ around every heavy atom. A cutoff of 5 Å for $r_t$ was used based on the typical upper distance between nuclei exhibiting a measurable NOE. Next, a frequency matrix $M$ was created, whose elements $M_{ij}$ are the relative amount of time that residues and $j$ spend closer than $r_t$ during the MD simulation. Upon computing the matrix $M$ for the WT system and for the K810A, K848A, and K855A mutants, the EP score per residue, $EP_i^x$, was calculated as follows:

$$EP_i^x = \sum_{j=1}^N \left[ M_{ij}^x - M_{ij}^{WT} \right],$$

where $x$ refers to the K810A, K848A, or K855A mutants and $N$ is the number of amino acids in HNH. For every mutant $x$, the total strength of EP (i.e., the total EP, $T_{EP}^x$) was defined as:

$$T_{EP}^x = \sum_{i=1}^N S_i^x .$$

where:

$$S_i^x = EP_i^x \quad if \quad EP_i^x > \langle EP^x \rangle_i + \sigma \left( EP^x \right)$$
$$S_i^x = 0 \quad if \quad EP_i^x < \langle EP^x \rangle_i + \sigma \left( EP^x \right) ,$$

with the angular brackets representing the ensemble average and $\sigma \left( EP^x \right)$ the standard deviation along the simulations. According to the above, the EP score is a quantitative measure of the total heavy atom contact change for every residue, occurring as a consequence of the different mutations. This quantity thereby measures the overall change in chemical environment and can be qualitatively compared with the composite NMR chemical shifts, which are a direct experimental reporter of the local environment.

## Communication pathway analysis

To describe the allosteric pathway of communication, information theory and network analysis were applied to the analysis of μs-length simulations (*Sethi et al., 2009*). First, a generalized correlations (*GC*) analysis (*Lange and Grubmüller, 2006*) was carried out to compute the overall correlations between Cα atoms of the HNH domain. This analysis quantifies the system's correlations based on the mutual information (*MI*) (*Lange and Grubmüller, 2006*) between two variables $x_i$ and $x_j$ (i.e., position vectors for the Cα atoms and $j$):

$$MI \left[ x_i, \ x_j \right] = H \left[ x_i \right] + H \left[ x_j \right] - H \left[ x_i, \ x_j \right]$$

where $H \left[ x_i \right]$ and $H \left[ x_j \right]$ are the marginal Shannon entropies and $H \left[ x_i, x_j \right]$ is the joint entropy, providing a link between motion correlations and information content (details are reported in S1. Supplementary materials and methods). The $MI$ can be converted into normalized $GC$, ranging from 0 (independent variables) to 1 (fully correlated variables):

$$GC_{ij} \left[ x_i, \ x_j \right] = \left\{ 1 - e^{-2MI \left[ x_i, \ x_j \right]/d} \right\}^{-1/2}$$

where $d = 3$ is the dimensionality of $x_i$ and $x_j$. The $GC$ values were used to build a network model, in which each residue was represented as a node connected by edges (*Sethi et al., 2009*). The lengths of the edges were weighted using the $GC$, with the weight ($w_{ij}$) of the edge connecting nodes and $j$ being:

$$w_{ij} = -logGC_{ij}$$

Hence, highly correlated pairs of residues are associated with efficient links for information transfer. Each node pair was connected by an edge if the residues involved spent at least 75% of the simulation time within 5 Å. This threshold was carefully optimized in our previous studies (*Palermo et al., 2017*; *East et al., 2020a*).

To determine the major channels of information flow, we employed the optimal path search introduced by Dijkstra in our network models (*Bowerman and Wereszczynski, 2016*). The resulting pathways were composed by single-edge steps that maximize the total correlation (and optimize the momentum transport) between the signal 'source' and 'sink' amino acids. We studied the communication pathways that traverse HNH from residues that connect REC (residues 789 and 794; sources) to RuvC (residues 841 and 858; sinks), and thereby transfer the information of DNA binding from the recognition region to the cleavage sites (*Sternberg et al., 2015*; *Chen et al., 2017*; *Dagdas et al., 2017*; *Palermo et al., 2017*; *Palermo et al., 2018*). The resulting routes maximize the dynamical crosstalk between the source and sink, serving as optimal communication channels. To account for the contribution of the most likely suboptimal pathways, the 10 shortest pathways for each source and sink pairs were computed, accumulated, and plotted on the 3D structure of HNH in its isolated form and within full-length Cas9.

## Community network analysis

To structure the allosteric network and visualize how it is altered by the K810A, K848A, and K855A mutations, we performed community network analysis (*Sethi et al., 2009*). The dynamical network models described above were divided into local substructures – i.e., communities – composed of groups of nodes in which the network connections are dense but between which they are sparse. Here, community network analysis has been based on GCs, since previous studies have shown to more comprehensively describe allosteric networks (*Palermo et al., 2017*; *East et al., 2020a*; *Melo et al., 2020*; *Saltalamacchia et al., 2020*). To structure the communities, the Girvan-Newman graph-partitioning approach (*Girvan and Newman, 2002*) was employed, using the EB as partitioning criterion (see Appendix 1). EB is defined as the number of shortest pathways that cross the edge and are computed using the Floyd-Warshall algorithm (*Floyd, 1962*), thereby accounting for the number of times an edge acts as a bridge in the communication flow between nodes of the network. The total EB between couples of communities (i.e., the sum of the EB of all edges connecting two communities) is a measure of their communication strength. Here, the total EB between couples of communities was used to quantify the mutation-induced changes in the communication flowing through HNH. For each mutant (K855A, K810A, and K484A), the mutation-induced EB change (ΔEB) was computed as a difference between the EB of the mutant and the WT system. To enable a proper comparison and to assess how the K-*to*-A mutations altered the WT communication, the communities of the WT system were used as a reference. Normalized ΔEB ranges from negative (–1 < 0) to positive (0 < 1) values, indicating loss or gain in communication, respectively. To further evaluate the dynamic exchange between communities, we analyzed the NMR CPMG relaxation dispersion data in the context of the community networks (*Lisi et al., 2017*). In this analysis, the HNH communities identified through community network analysis were used as a reference, and the dynamic exchange among them was derived from CPMG relaxation dispersion experiments. A correlation matrix was produced by multiplying the number of residues in each community with that of the adjacent communities displaying CPMG relaxation dispersion (*Supplementary file 1*). The resulting correlation matrices for the WT and each mutant were normalized based on the total number of CPMG active residues and plotted as a community network (*Figure 5—figure supplement 1*). This analysis enabled us to evaluate the exchange of dynamics (experimentally measured) between communities. Details are in S1. Supplementary materials and methods and S2. Supplementary results.

## Acknowledgements

We thank Amelia Palermo for useful insights on circular correlation networks and data visualization. This material is based upon work supported by the National Institute of Health under Grant No R01GM136815 (awarded to VSB, GP, and GPL) and Grant No R01GM141329 (awarded to GP). This work was also funded by the National Science Foundation under Grant No CHE-1905374 (awarded to GP). Computer time for MD has been awarded by XSEDE under Grant No TG-MCB160059 and by NERSC under Grant No M3807 (to GP).

## Additional information

### Funding

| Funder | Grant reference number | Author |
|---|---|---|
| National Institutes of Health | R01GM141329 | Giulia Palermo |
| National Science Foundation | CHE-1905374 | Giulia Palermo |
| National Institutes of Health | R01GM136815 | Victor S Batista George P Lisi Giulia Palermo |

The funders had no role in study design, data collection and interpretation, or the decision to submit the work for publication.

## Author contributions
Lukasz Nierzwicki, Kyle W East, Data curation, Formal analysis, Writing – review and editing; Uriel N Morzan, Formal analysis, Methodology; Pablo R Arantes, Data curation, Formal analysis, Methodology; Victor S Batista, Funding acquisition; George P Lisi, Conceptualization, Funding acquisition, Project administration, Supervision, Writing - original draft, Writing – review and editing; Giulia Palermo, Conceptualization, Funding acquisition, Investigation, Project administration, Supervision, Writing - original draft, Writing – review and editing

## Author ORCIDs
George P Lisi ⬚ http://orcid.org/0000-0001-8878-5655
Giulia Palermo ⬚ http://orcid.org/0000-0003-1404-8737

## Decision letter and Author response
Decision letter https://doi.org/10.7554/eLife.73601.sa1
Author response https://doi.org/10.7554/eLife.73601.sa2

# Additional files

## Supplementary files
• Supplementary file 1. Additonal MD simulationa dn NMR data pertaining to the allosteric role of specificity enhancement mutations in HNH.

• Transparent reporting form

## Data availability
Analysis codes and script files can be downloaded from Github: https://github.com/palermolab Resonance assignments for the HNH structure are available at bmrb.io under BMRB entry 27949.

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

# Appendix 1

## S1. Supplementary materials and methods

### S1.1. Structural models for molecular simulations

MD simulations have been performed on the full-length CRISPR-Cas9 system and on the isolated form of the HNH domain. The X-ray structures of the *Streptococcus pyogenes* (Sp) CRISPR-Cas9 (PDB: 5F9R, at 3.40 Å) *Jiang and Doudna, 2017* and of the isolated HNH domain (PDB: 6O56, at 1.90 Å resolution) *East et al., 2020a* have been used as models. The HNH domain (residues 775–908) superposes well in the two structures, displaying a root mean square deviation (RMSD) difference for the Cα below 1 Å (i.e., 0.92 Å). Both model systems have been object of MD as WT and introducing the three single-point mutations K810A, K848A, and K855A *Slaymaker et al., 2016*, resulting in eight simulation systems. The systems composed of the isolated HNH were solvated in an ~80 Å × ~70 Å × ~65 Å box of ~34,000 atoms, while the full-length CRISPR-Cas9 complexes were solvated in an ~180 Å × ~117 Å × ~140 Å box of ~340,000 total atoms. Three additional systems, including a longer non-target DNA strand, were also built to simulate the triple mutants K810A-K1003A-R1060A (eSpCas9 1.0) and K848A-K1003A-R1060A (eSpCas9 1.1) *Slaymaker et al., 2016*, as well as the WT Cas9. In these systems, the DNA non-target strand reaches the location of the K1003 and R1060 residues, enabling to assess the effect of their alanine mutation on the protein-nucleic acid interactions. For these systems, the cryo-EM structure EMD-8236 (5Y36.pdb, at 5.20 Å resolution) *Huai et al., 2017* was used, since it provides structural information of the terminal bases of the DNA non-target strand. These three additional systems comprised a total of ~412,000 atoms, solvated in an ~205 Å × ~167 Å × ~126 Å box. In all systems, counterions were added to provide physiological ionic strength.

### S1.2. MD simulations

The above-mentioned model systems were equilibrated and production runs were performed using the new AMBER ff99SBnmr2 force field *Yu et al., 2020*, which improves the consistency of the backbone conformational ensemble with NMR experiments, was used for the protein. Nucleic acids were described, including the ff99bsc0 corrections for DNA *Pérez et al., 2007* and the ff99bsc0+ $\chi$ OL3 corrections for RNA *Zgarbová et al., 2011*; *Banáš et al., 2010*. The TIP3P model was used for water *Jorgensen et al., 1983*. Simulations were performed in the NPT ensemble with temperature held at 310 K using the Bussi velocity rescaling thermostat *Bussi et al., 2007*. The pressure was held at 1 bar with the Parrinello-Rahman barostat *Parrinello and Rahman, 1981*. The particle mesh Ewald method was used to compute long-range electrostatic interactions with a cutoff radius of 1 nm. All bonds involving hydrogen were restrained using the LINCS algorithm. The equation of motion was integrated using a time step of 2 fs. An ~1.2 µs-long trajectory was collected in three replicas for the WT CRISPR-Cas9 system and for each of the K845A, K855A, and H855A variants. The isolated HNH domain was also simulated in three replicas of ~1.2 µs each as WT and introducing the three K-to-A points. This resulted in ~3.6 µs of MD for each simulated system and a total of ~14.4 µs of MD for the full-length CRISPR-Cas9 and the isolated HNH domain. This simulation length (in three replicates) was motivated by our previous work *Palermo et al., 2017*; *Palermo et al., 2018*; *Jiang et al., 2016*; *Palermo et al., 2016*; *East et al., 2020a*, showing that it provides a solid statistical ensemble for the purpose of the analysis of the allosteric mechanism (described below). Analysis of the results has been performed upon discarding the first ~200 ns of MD, to enable proper equilibration and a fair comparison. Data are reported for the overall ensemble in the main text and for the separated replicas in Appendix 1. The enlarged model systems of the WT Cas9, SpCas9 1.0, and SpCas9 1.1 were also simulated for ~1.2 µs, each. All simulations were conducted using Gromacs (v. 2018.3) *Van Der Spoel et al., 2005*.

### S1.3. EP score

To determine the structural perturbation induced by the three K-to-A mutations from MD simulations, we introduced an EP score measure. The EP score is a measure of the mutation-induced dynamic perturbation suffered by each residue in HNH given its local environment. The EP score analysis begins with the definition of a threshold radius $r_t$ around every heavy atom. A cutoff of 5 Å for $r_t$ has been used based on the typical upper distance between nuclei exhibiting a measurable NOE. Next, a frequency matrix $M$ was created, whose elements $M_{ij}$ are the relative amount of time that residues $i$ and $j$ spend closer than $r_t$ during the MD simulation. Upon computing the matrix $M$ for the WT system and for the K810A, K848A, and K855A mutants, the EP score per residue, $EP_i^x$, was calculated as follows:

$$EP_i^x = \sum_{j=1}^{N} [M_{ij}^x - M_{ij}^{WT}],$$

where $x$ refers to the K810A, K848A, or K855A mutants and $N$ is the number of amino acids in HNH. Finally, for every mutant $x$, the total strength of chemical EP (i.e., the total EP, $T_{EP}^x$) was defined as:

$$T_{EP}^x = \sum_{i=1}^{N} S_i^x.$$

where:

$$S_i^x = EP_i^x \quad if \quad EP_i^x > \langle EP^x \rangle_i + \sigma(EP^x)$$
$$S_i^x = 0 \quad if \quad EP_i^x < \langle EP^x \rangle_i + \sigma(EP^x),$$

with the angular brackets representing the ensemble average and σ(EP^$x$ ) the standard deviation along the simulations. According to the above, the EP score is a quantitative measure of the total heavy atom contact change for every residue, occurring as a consequence of the different mutations. This quantity thereby measures the overall change in chemical environment and can be qualitatively compared with the composite NMR chemical shifts, which are a direct experimental reporter of the local environment.

## S1.4. Community network analysis

The allosteric mechanism of communication was examined by combining information theory and community network analysis **Sethi et al., 2009**. In this analysis, biomolecular systems are described as a network of amino acids residues (i.e., nodes of the network) connected by edges (residue pair connection). To account for the information exchange between amino acids (represented by their Cα atoms), the length of the edges connecting nodes is related to their motion correlations. Here, correlated motions between residue pairs are computed using the GC method by Lange and Grubmüller (**Lange and Grubmüller, 2006**), which quantifies the system's correlations based on MI. In this method, two variables ($x_i, x_j$) can be considered correlated when their joint probability distribution, , is smaller than the product of their marginal distributions, . The MI is a measure of the degree of correlation between and defined as function of and according to:

$$MI[x_i, x_j] = \int \int p(x_i, x_j) ln \frac{p(x_i, x_j)}{p(x_i) \cdot p(x_j)} dx_i dx_j$$

Notably, MI is closely related to the definition of the Shannon entropy, $H[x]$, that is, the expectation value of a random variable $x$, having a probability distribution $p(x_i)$

$$H[x] = \int p(x) \, ln \, p(x) dx$$

and it can be thus computed as:

$$MI[x_i, x_j] = H[x_i] + H[x_j] - H[x_i, x_j]$$

where $H[x_i]$ and $H[x_j]$ are the marginal Shannon entropies, and $H[x_i, x_j]$ is the joint entropy. The g_correlation tool **Lange and Grubmüller, 2006** implemented in Gromacs 3.350 was used to calculate the marginal entropies $H[x_i]$ and $H[x_j]$ and the joint entropy $H[x_i\_x_j]$ by means of the $k$-nearest neighbor distances algorithm **Floyd, 1962**, applied to the atomic positions' fluctuations from MD simulations. Since MI varies from 0 to +∞, it can be converted into normalized generalized correlation coefficients (GC), ranging from 0 (independent variables) to 1 (fully correlated variables), are defined as:

$$GC_{ij}[x_i, x_j] = \{1 - e^{-2MI[x_i, x_j]/d}\}^{-1/2}$$

where $d$=3, the dimensionality of $x_i$ and $x_j$. With respect to the more traditional Pearson's coefficients analysis, GC analysis has the advantage of capturing correlations independently on the relative orientation of the atomic fluctuations, while also being able to capture non-linear

correlations. Network analysis was originally introduced by Luthney-Schulten building on Pearson's coefficients *Sethi et al., 2009*. More recently, the use of GCs has shown to more comprehensively describe allosteric networks *Palermo et al., 2017*; *East et al., 2020a*; *Melo et al., 2020*; *Saltalamacchia et al., 2020*. Based on GCs, a network of nodes (i.e., amino acid Cα) connected by edges is built, with the weight ($w\_ij$) of the edge connecting nodes $i$ and $j$ computed as:

$$W_{ij} = -logCG_{ij}$$

Two nodes are considered connected if any heavy atom of the two residues is within 5 Å of each other (i.e., distance cutoff) for at least the 75% of the simulation time (i.e., frame cutoff). This threshold was carefully optimized in our previous studies *Palermo et al., 2017*; *East et al., 2020a*, based on the estimation of the community repartition difference (CRD), defined as:

$$CRD(c_1, c_2) = 1 - \frac{\sum_{n_i,n_j} z(n_i,n_j,c_1)z(n_i,n_j,c_2)}{\sum_{n_i,n_j} z(n_i,n_j,c_1)}$$

where $z(n_i, n_j, c_i)$ is defined as 1 if nodes $n_i$ and $n_j$ belong to the same community in a given partition $c\_i$ (i.e., the community structure) and 0 otherwise. CRD is an estimate of the similarities between different network partitions, as from community structures obtained with different cutoff values. Our previous work considering multiple runs of the WT Cas9 estimated converged values of CRD for the two selected cutoff parameters *Palermo et al., 2017*; *East et al., 2020a*. The resulting 'weighted graph' defines the system as a dynamical network, with information on the critical nodes that are important for the communication within the complex. In the weighted network, a set of 'communities' can be identified. Communities are groups of nodes in which the network connections are dense but between which they are sparse. These local substructures can be obtained with the Girvan-Newman algorithm *Girvan and Newman, 2002*, which is a divisive algorithm that uses the 'EB' partitioning criterion. The EB are defined as the number of shortest pathways that cross the edge, and are computed using the Floyd-Warshall algorithm (*Floyd, 1962*), which sums the lengths ($w_{ij}$) of all edges involved in different paths of nodes connecting two distant residues and identifies the pathway displaying the shortest total length. EB accounts for the number of times an edge acts as a bridge in the communication flow between nodes of the network. EB is a parameter that favors edges that inter-connect communities and disfavors edges that lie within communities. High EB thereby associates with pairs of residues that are important for the communication flow within the weighted network. The Girvan-Newman algorithm for finding communities implements an iterative process, where the edge with the highest EB is removed from the network and the betweenness of the remaining edges is recalculated, with communities being progressively isolated up to the point where each node will represent a community. The optimal division of the network has to be obtained in such a way that each community contain nodes that are highly intra-connected while different communities are poorly inter-connected through few critical nodes. The parameter of 'modularity', $Q$, measures the strength (or the quality) of the community structure and it is used for determining the optimal division of the network. $Q$ represents the difference in probability of intra- and inter-community connections for a given network division and is defined as:

$$Q = \sum{}_i(e_{ii} - a_i^2)$$

where $e_{ij}$ is the fraction of edges that links nodes in community $i$ to nodes in community $j$, while $a_i = \sum{}_j e_{ij}$ is the fraction of edges that connects to nodes in community $i$. The modularity value falls in the range of 0–1, with larger values indicating higher community structure quality. The optimum community structures here have the highest modularity $Q = 0.7$. This cutoff was applied for the WT system and for each of the three K-to-A mutants, in line with our previous studies of the allosteric mechanism in Cas9 *Palermo et al., 2017*; *East et al., 2020a*. This value is also consistent with modularity values found for other biomolecular systems (0.4–0.8) *Saltalamacchia et al., 2020*.

## S.1.5. Communication pathway analysis
To determine the major channels of information flow, we employed the optimal path search introduced by Dijkstra in our network models *Bowerman and Wereszczynski, 2016*. The algorithm

uses the generalized correlation coefficients ($GC_{ij}$) as a metrics to define the iterative optimization problem. It finds the roads, composed by the $w_{(ij)}^0$ inter-node connections, which minimize the total distance (and maximize the correlation) between 'source' and 'sink' nodes. We considered residues that connect the DNA recognition region REC (residues 789 and 794) to the RuvC catalytic domain (residues 841 and 858) as source and sink of the communication (full details are in the main text). The resulting routes maximizing the dynamical crosstalk between source and sink, including the 10 shortest pathways, are plotted on the 3D structure of HNH within the full-length Cas9 and in its isolated form (*Appendix 1—figures 1 and 2*). The Dijkstra algorithm was also applied building on Pearson's coefficients, revealing the consistency of the computed pathways (*Appendix 1—figures 3 and 4*).

### S1.6. Mutation induced ΔEB

High EB are associated with pair of residues that are important communication flows within the weighted network. The total EB between couples of communities (i.e., the sum of the EB of all edges connecting two communities) is a measure of the communication strength between communities. Here, the total EB between couples of communities has been used to quantify the mutation-induced changes in the communication flowing through HNH. For each mutant (K855A, K810A, and K484A), the mutation-induced EB change (ΔEB) has been computed for each couple of communities as a difference between the EB of the mutant and the WT system. To enable a proper comparison and, most importantly, to assess how the three K-to-A mutations alter the WT communication, the communities of the WT system have been used as a reference. Community network analysis identified seven communities for both the HNH domain within the full-length Cas9 and in its isolated form (*Appendix 1—figure 5*). Three communities hold most of the residues that compose the allosteric pathway in the WT Cas9 and are thereby 'allosteric' communities (A1, yellow, A2 cyan, and A3 purple). The 'non-allosteric' communities (NA1 orange, NA2 tan, NA3 red, and NA4 black) include only few allosteric residues. These findings were consistent in the simulation replica and were used as a reference for the calculation of the ΔEB for the K855A, K810A, and K484A mutants. In detail, the EB between couples of communities of the HNH domain have been computed for the WT system and for the three mutants and reported using two-by-two matrices (Appendix figures 6, 7) and community network plots (*Appendix 1—figure 8*). The averaged (between three replicas) EB have been used to compute the ΔEB as a difference between the between the EB of the mutant and the WT system. As a result, negative values of ΔEB indicate loss of communication between couples of communities, while positive values of ΔEB indicate gain in communication induced by the mutations. The ΔEB for the three mutants have been normalized and plotted using circular graphs, where the communities are disposed in a circle and connected using links, whose thickness is proportional to the ΔEB and is color-coded using a red (negative, from −1 to 0) to blue (positive, from 0 to 1) scale (*Appendix 1—figure 8*). Data are reported for the three K-to-A mutants considering HNH in both its isolated form and within the full-length Cas9.

## S2. Supplementary results

### S2.1. Molecular simulations of the triple mutants

We performed three additional μs-length MD simulations of the WT Cas9 and its triple mutants K810A-K1003A-R1060A (viz., eSpCas9 1.0) and K848A-K1003A-R1060A (viz., eSpCas9 1.1) *Slaymaker et al., 2016*. We considered an enlarged model system of CRISPR-Cas9, including a longer DNA non-target strand reaching the K1003 and R1060 residues (*Figure 7A*). We analyzed the fluctuations of the DNA bases composing the 20 nucleotides (20-nt) segment that locates within the RuvC domain, which holds the K1003 and R1060 residues. For each nucleotide, the RMSD of the heavy atoms with respect to the initial positions have been computed along the simulations and plotted using box plots (*Figure 7B*). As a result, the eSpCas9 1.0 and eSpCas9 1.1 systems, which lack the positively charged K1003 and R1060, display an increase in the fluctuations of the DNA bases, compared to the WT Cas9. This increase is remarkable from the middle to the end of the 20-nt segment, which is closer to the K1003A and R1060A mutations. These increased fluctuations indicate high flexibility of the distal DNA non-target strand, resulting from the loss of electrostatic interactions between the DNA non-target strand and the eSpCas9 1.0 and eSpCas9 1.1 proteins *Slaymaker et al., 2016*. Such loss of interactions could possibly facilitate the re-hybridization of DNA, reducing off-target cleavages as previously suggested *Singh et al., 2016*; *Singh et al., 2018*. An in-depth investigation of the eSpCas9 1.0 and eSpCas9 1.1 systems at the molecular level is currently ongoing in our laboratories to gain further insights on their mechanism of action.

**FULL LENGHT Cas9**

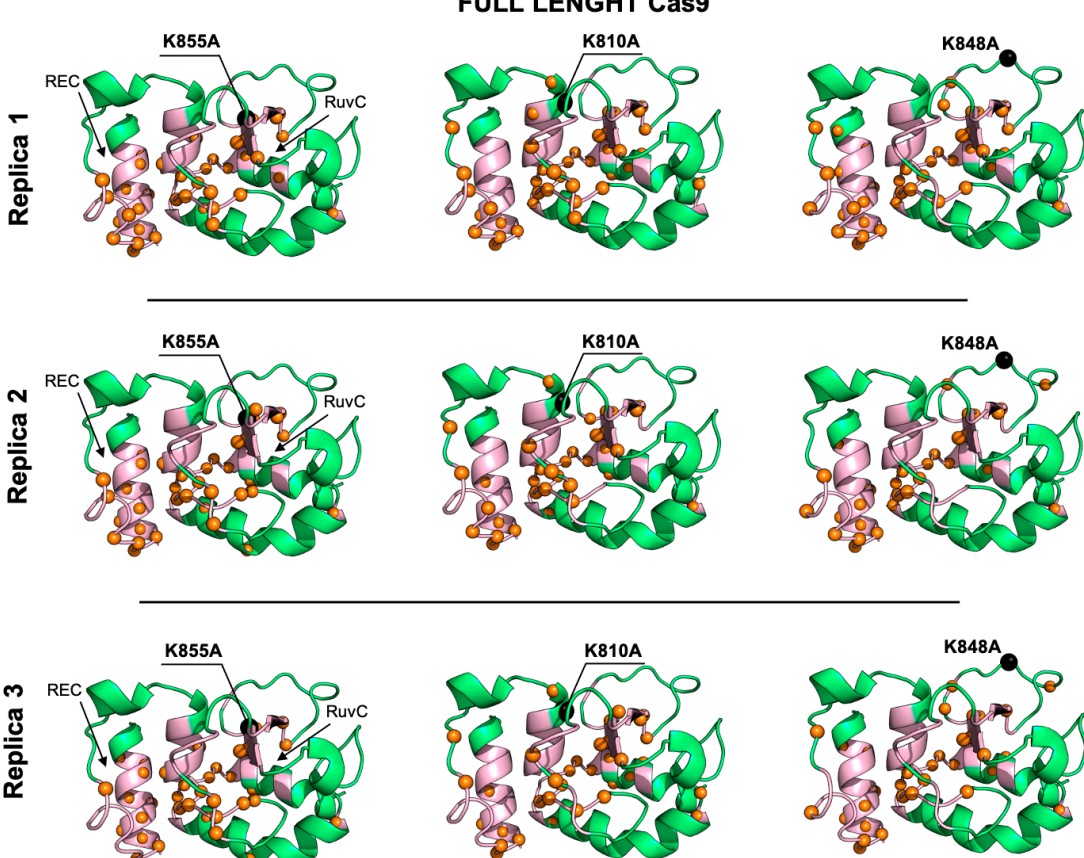

**Appendix 1—figure 1.** Allosteric dynamical pathways across the HNH domain for the K855A, K810A, K848A mutants, computed from µs-length molecular dynamics (MD) of the full-length Cas9 (data are shown for three simulation replicas). The residue-to-residue dynamical pathways optimizing the momentum transport from REC to RuvC are shown using orange spheres. The HNH domain (green) is shown on side view. The wild-type (WT) allosteric pathway (pink ribbons), previously identified through µs-length MD and Carr-Purcell-Meiboom-Gill (CPMG) relaxation dispersion *East et al., 2020a*, is also shown for reference. The K855A, K810A, and K848A mutations are shown using black spheres.

**ISOLATED HNH**

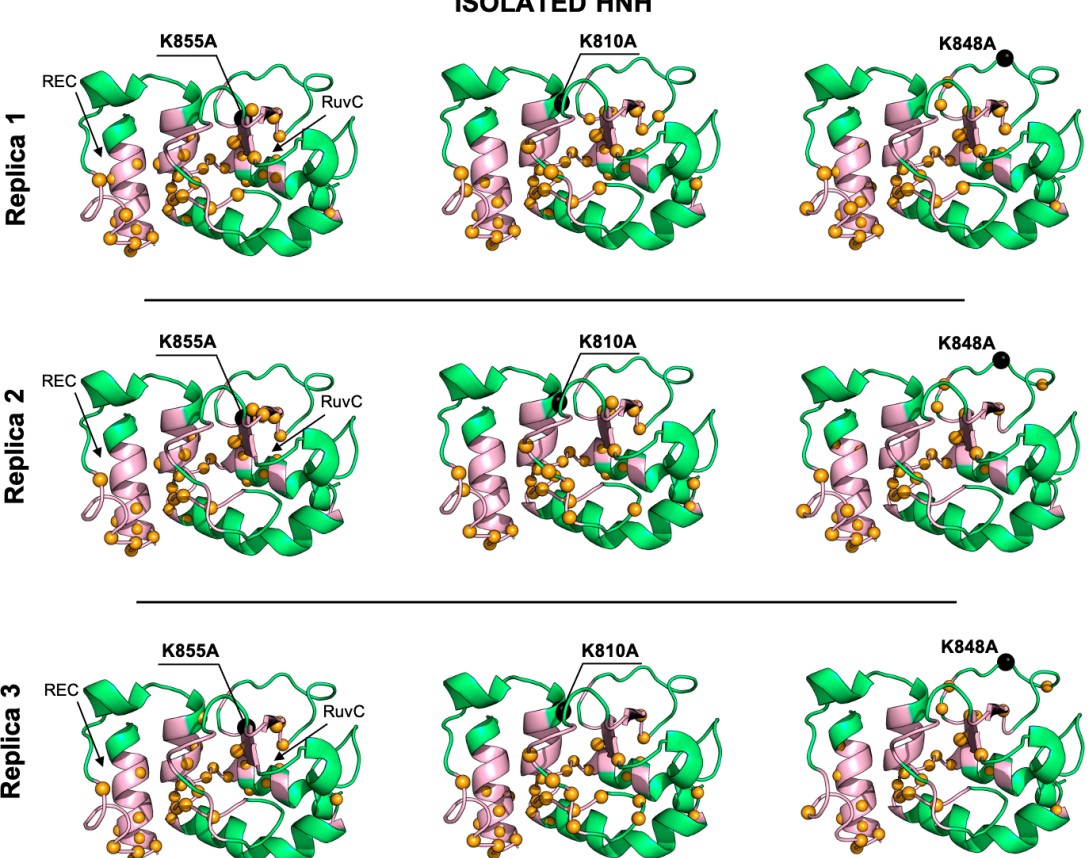

**Appendix 1—figure 2.** Allosteric dynamical pathways across the HNH domain for the K855A, K810A, K848A mutants, computed from μs-length molecular dynamics (MD) of the isolated HNH domain (data are shown for three simulation replicas). The residue-to-residue dynamical pathways optimizing the momentum transport from REC to RuvC are shown using yellow spheres. The HNH domain (green) is shown on side view. The wild-type (WT) allosteric pathway (pink ribbons), previously identified through μs-length MD and Carr-Purcell-Meiboom-Gill (CPMG) relaxation dispersion (*East et al., 2020a*), is also shown for reference. The K855A, K810A, and K848A mutations are shown using black spheres.

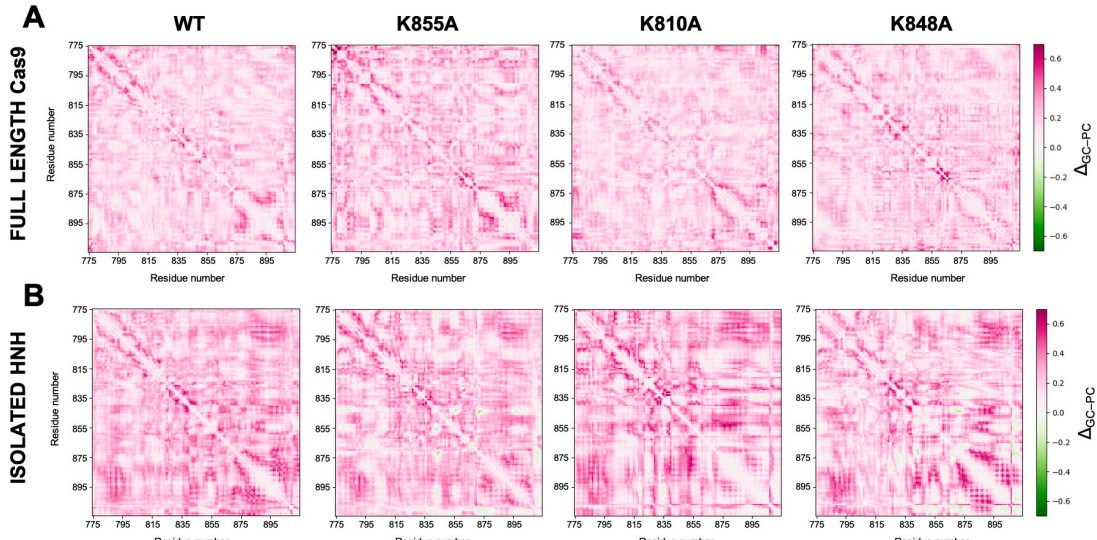

**Appendix 1—figure 3.** Comparison of Lange and Grubmüller's generalized correlations (GCs) and Pearson's correlations (PCs) for allosteric pathways analysis. Difference matrices between the GCs and PCs ($\Delta_{GC-PC}$), computed for the wild-type (WT) HNH and the K855A, K810A, and K848A mutants in the full-length Cas9 (A) and in the isolated HNH domain (B). The $\Delta_{GC-PC}$ matrices have been computed considering the absolute values of PCs, and on the overall MD ensemble (i.e., three replicas of ~1 µs for each system). $\Delta_{GC-PC}$ matrices are plotted according to the color-code of the scale on the right: from purple (GCs higher than PCs) to green (GCs lower than PCs) through white (GCs similar to PCs). This analysis shows that the GCs are higher than the absolute PCs values, indicating that the GC method can capture more correlations.

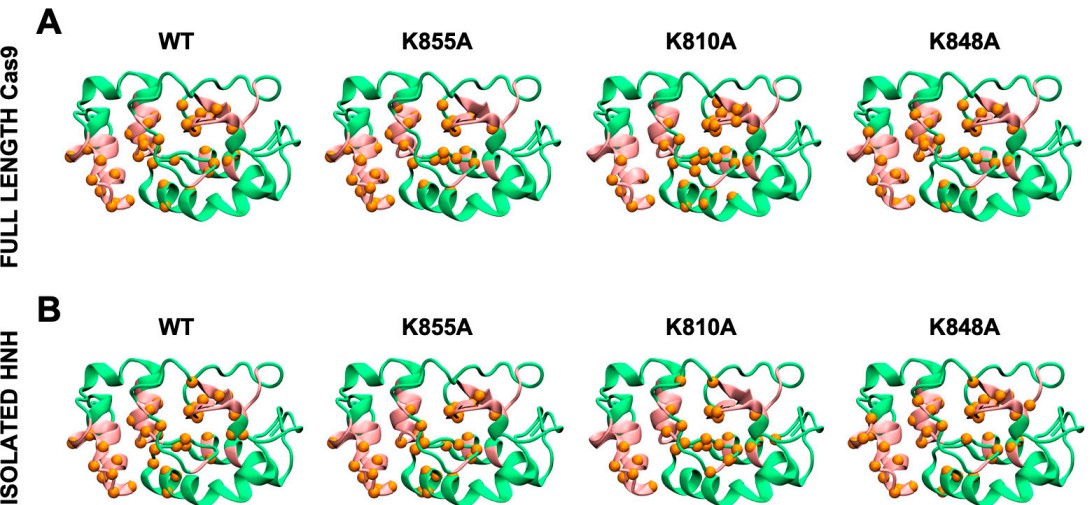

**Appendix 1—figure 4.** Allosteric pathways derived from Pearson's coefficients (PCs), computed for the wild-type (WT) HNH and the K855A, K810A, and K848A mutants in the full-length Cas9 (A) and in the isolated HNH domain (B). Residues composing the HNH allosteric pathways are shown as orange spheres. The WT allosteric pathway previously identified through molecular dynamics MD and NMR (pink ribbons) (*East et al., 2020a*) is also shown as a reference. For each system, the analysis of the allosteric pathways has been performed on the overall MD ensemble (i.e., three replicas of ~1 µs each), revealing a qualitative agreement with the pathways derived from GCs (*Appendix 1—figures 1 and 2*), and confirming that the allosteric dynamic signaling is preserved in the Cas9 mutants. This indicates that both methods convey in describing the systems' correlations, yet the GC method is a more complete estimator of the overall coupled motions.

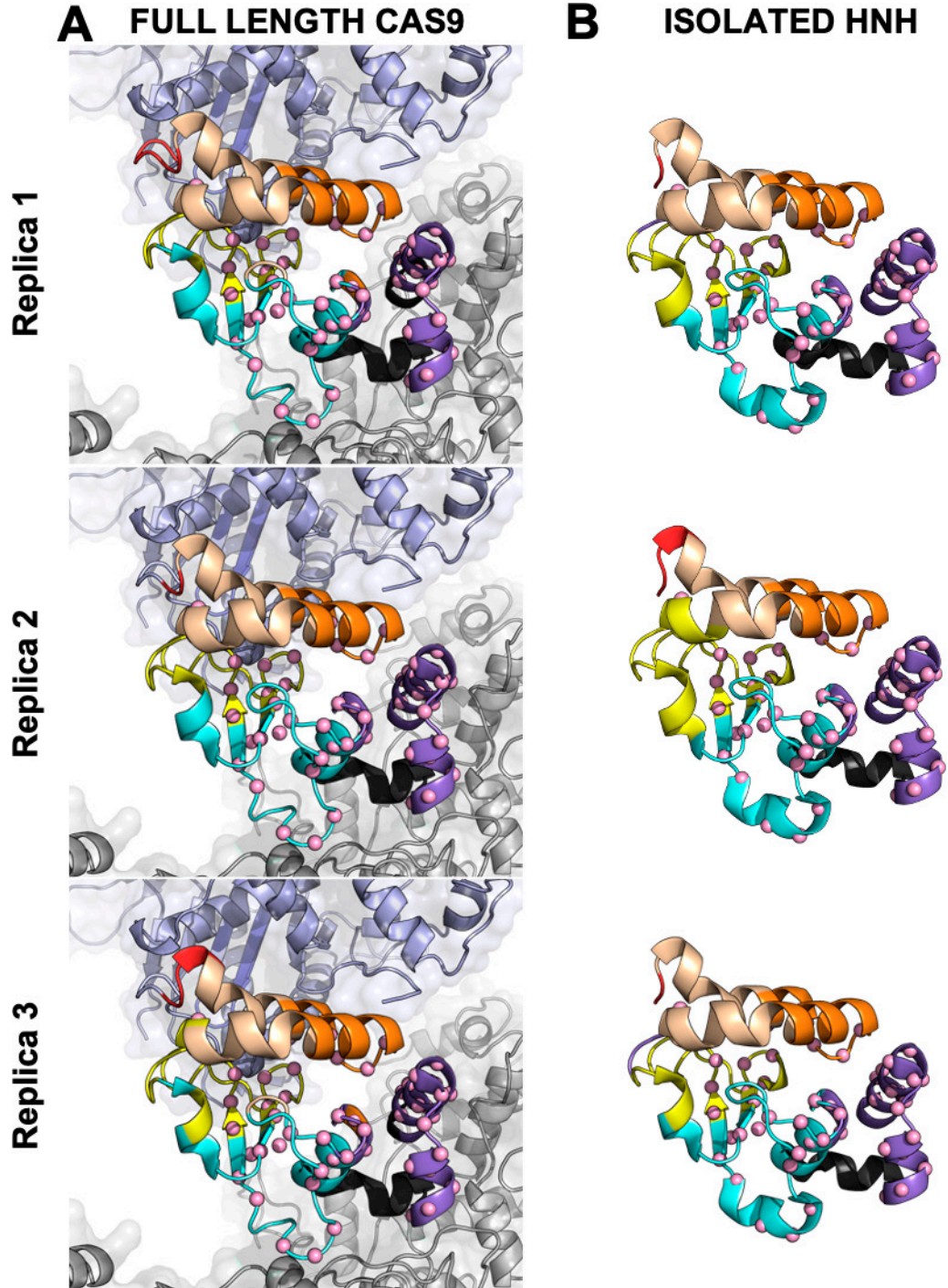

**Appendix 1—figure 5.** Close-up view of the HNH domain within the wild-type (WT) CRISPR-Cas9 (A) and in its isolated form (B). Seven communities of synchronized dynamics are shown using different colors. Data are reported for three simulation replicas. Three communities are allosteric (A1 yellow, A2 cyan, and A3 purple), since they hold most of the residues that compose the allosteric pathway (shown as spheres). The non-allosteric communities (NA1 orange, NA2 tan, NA3 red, NA4 black) include a few allosteric residues.

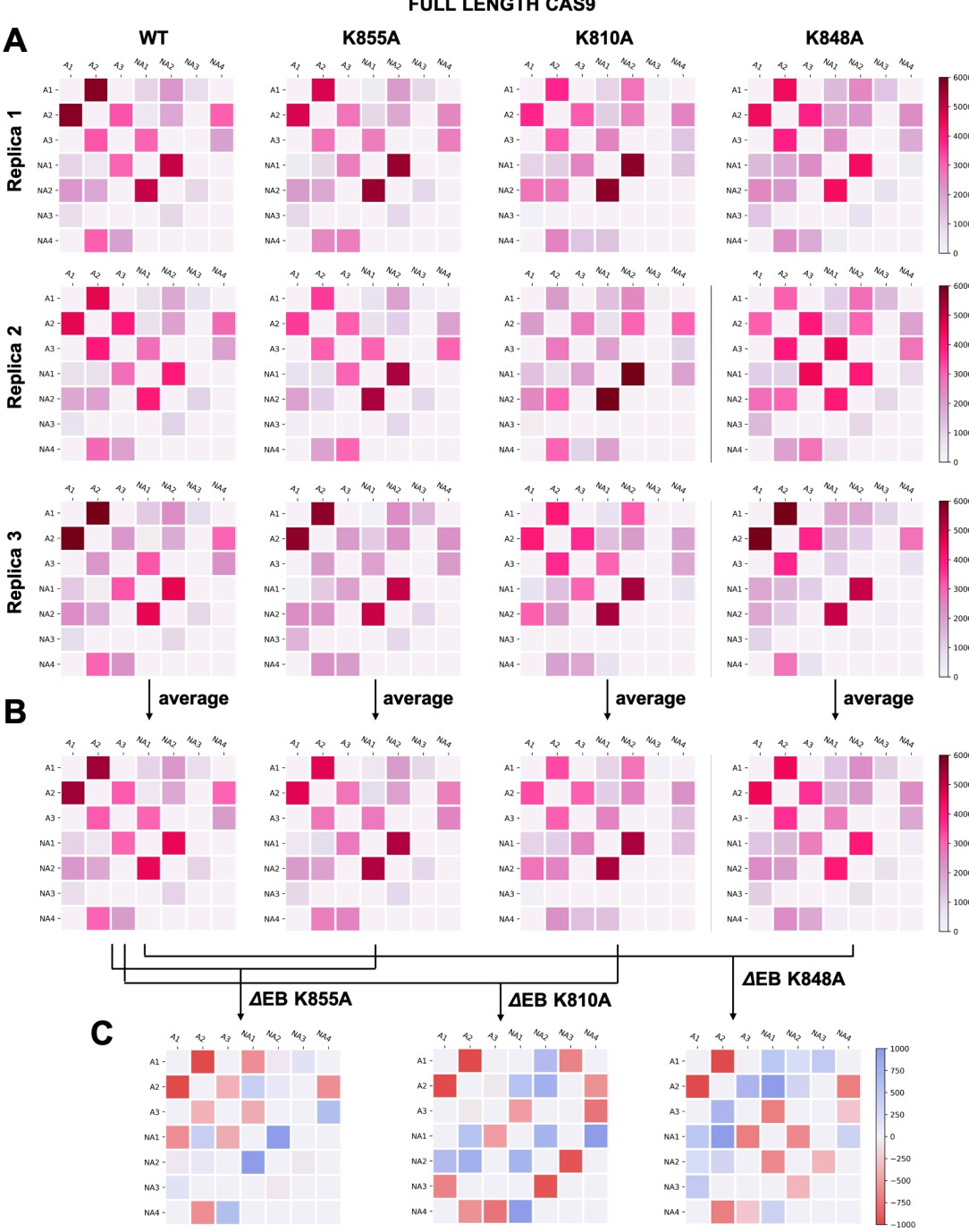

**Appendix 1—figure 6.** (A–B) Two-by-two matrices of the edge betweennesses (EB), computed for the wild-type (WT) system and for the K855A, K810A, and K848A mutants. The EB are computed for each couple of communities of HNH in the full-length Cas9 and are plotted according to the scale on the right. Data are reported for three simulation replicas (A), and for the average between them (B). (C) Two-by-two matrices of the mutation-induced EB change (ΔEB), computed as a difference between the averaged EB of the mutants and of the WT system. Values of ΔEB are plotted according to the scale on the right. The ΔEBs have also been plotted as circular graphs (**Appendix 1—figure 8A**).

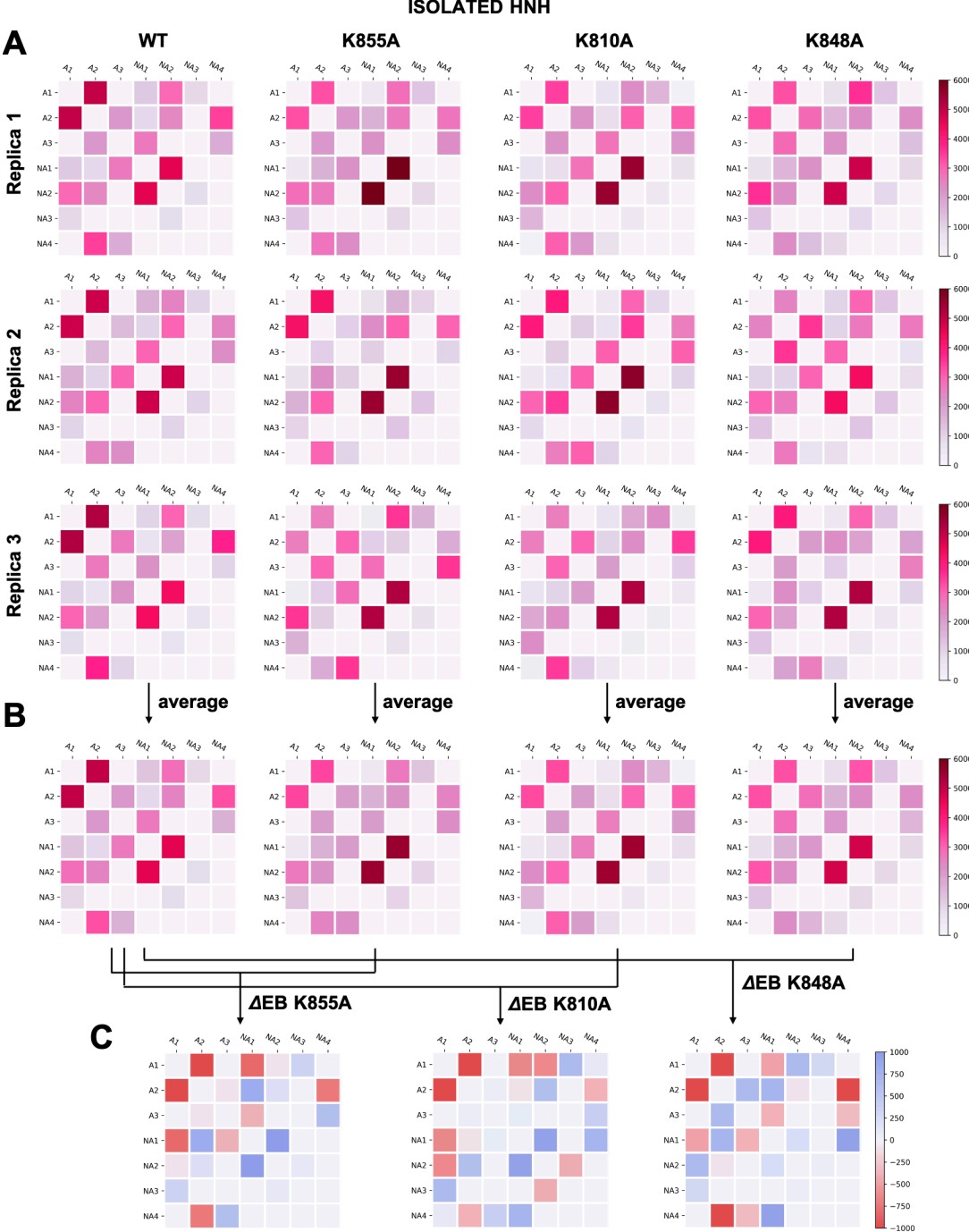

**Appendix 1—figure 7.** (A–B) Two-by-two matrices of the edge betweennesses (EB), computed for the wild-type (WT) system and for the K855A, K810A, and K848A mutants. The EB are computed for each couple of communities of HNH in its isolated form and are plotted according to the scale on the right. Data are reported for three simulation replicas (A), and for the average between them (B). (C) Two-by-two matrices of the mutation-induced EB change (ΔEB), computed as a difference between the averaged EB of the mutants and of the WT system. Values of ΔEB are plotted according to the scale on the right. The ΔEBs have also been plotted as circular graphs (*Appendix 1—figure 8B*).

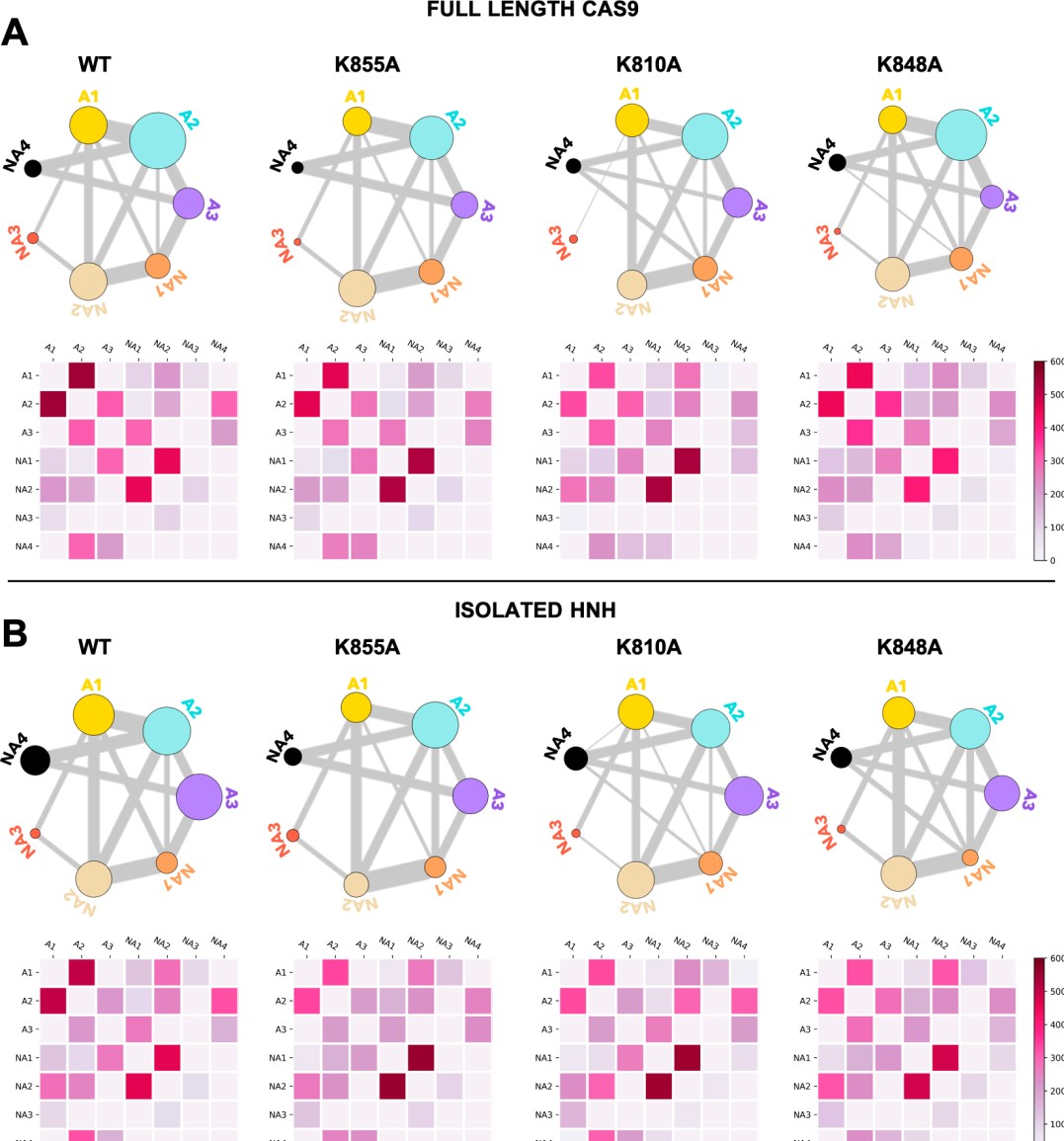

**Appendix 1—figure 8.** Community network plots (top) and corresponding two-by-two matrices (bottom) of the edge betweennesses (EB) computed for the wild-type (WT) HNH and for the K855A, K810A, and K848A mutants in the full-length Cas9 (A) and in the isolated HNH domain (B). Bonds connecting communities are a representation of their EB and hence of the communities' intercommunication strength. Data are reported for the average EB matrices in *Appendix 1—figures 6 and 7*. The EB matrices are plotted according to the scale on the right.

