## [Editor Report]

This paper presents an elegant and multidisciplinary study combining state-of-the-art NMR with computational modeling methods, to characterize the effects of mutations on the structure and allosteric communication within the CRISPR-Cas9 system. In revealing the link between the allosteric network in the protein and the increase in CRISPR-Cas9 specificity, this study carries important implications for the design of new gene editing tools.

---

## [Decision Letter]

**Decision letter after peer review:**

Thank you for submitting your article "Enhanced Specificity Mutations Perturb Allosteric Signaling in CRISPR-Cas9" for consideration by *eLife*. Your article has been reviewed by 3 peer reviewers, and the evaluation has been overseen by Rina Rosenzweig as Reviewing Editor and José Faraldo-Gómez as the Senior Editor. The following individual involved in review of your submission has agreed to reveal their identity: Ivaylo Ivanov (Reviewer #1).

Essential revisions:

1) One significant concern raised by several reviewers is the lack of a direct comparison between the NMR studies and the MD simulations. This should be addressed in the revised manuscript.

2) The authors performed relaxation measurements for fast dynamics, however, they did not calculate the order parameters for the protein backbone. These should therefore be calculated and the authors should clearly indicate how the order parameters and heteronuclear NOEs compare to the calculated values from the MD trajectories.

3) There is no mention of figures 4B and 4C in the manuscript. The results presented in these figures should be discussed, and protein regions showing slow and fast timescale dynamics should be clearly indicated.

4) CPMG data were collected at multiple fields and the data analyzed, yet the analysis of these results was not presented. The authors should provide the kex and pbs obtained for these data, as well as indicate the changes between the wild type and mutant proteins. Do all the mutants populate the same excited state (as gauged from δ omegas)? How does this fit with the MD?

5) The effect of the mutants on the micro-to-millisecond timescale dynamics should be discussed. Additionally, it is unclear how the dynamics or structural perturbations caused by these selected mutants are converted into the enzyme's increased or decreased specificity.

6) The authors state that the differences in the relaxation dispersion profiles are less than 1.5 Hz, indicating small changes in dynamics. A Plot showing the differences in the relaxation dispersion profiles (ΔRex) should be provided for all the proteins (WT and mutants) to support this claim.

7) The authors have chosen to use Grubmuller's generalized correlation to compute the weights on the nodes of the protein network. Grubmuller's generalized correlation captures both linear and non-linear correlations. Indeed, you could run the linearized version to distinguish the non-linear from linear correlations. Would the results have been different if Pearson correlation was used? Conversely, would there be key allosteric residues picked up by generalized correlation and not by linear correlation?

8) When using the Girvan-Newman method to partition the network graph into communities, it is possible that the different simulation ensembles for the K855A, K810A and K484A mutants could result in different numbers of residues per community. In that case, it becomes difficult to compare the changes in betweenness for the mutants as there is also an accompanying shift in residues between communities. Could the authors please confirm that this is not the case and that each community contains the exact same number of residues for the K855A, K810A and K484A mutants.

9) Did the authors employ a modularity cutoff for the Girvan-Newman method to control community subdivision? And if so, was the cutoff the same for each of the three mutant cases?

10) In the discussion, the authors refer to the synchronous motions that may be responsible for specificity. How did they deduce that the motions are synchronous? From MD simulations or the global fitting of the CPMG curves? Do motions need to be synchronous for effective allosteric communications?

11) The authors state that mutations can target sites identified in this study (hotspots) to improve CRISP-Cas9 function. Can the authors elaborate more on this point? How do they envision the mutations could tune the function of the complex?

12) Generally, there appear to be a number of grammatical and stylistic issues with the manuscript. Revisiting the writing could serve to improve the readability of the article.

*Reviewer #1 (Recommendations for the authors):*

My area of expertise is molecular modeling. Therefore, I will constrain my comments mostly to the molecular modeling and computational aspects of the manuscript. I would appreciate if the authors address the following points in the revised version:

1. The authors have chosen to use Grubmuller's generalized correlation to compute the weights on the nodes of the protein network. Grubmuller's generalized correlation captures both linear and non-linear correlations. Indeed, you could run the linearized version to distinguish the non-linear from linear correlations. Would the results have been different if Pearson correlation was used? Conversely, would there be key allosteric residues picked up by generalized correlation and not by linear correlation?

2. When using the Girvan-Newman method to partition the network graph into communities, it is possible that the different simulation ensembles for the K855A, K810A and K484A mutants could results in different numbers of residues per community. In that case, it becomes difficult to compare the changes in betweenness for the mutants as there is also an accompanying shift in residues between communities. Could the authors please confirm that this is not the case and that each community contains the exact same number of residues for the K855A, K810A and K484A mutants.

3. Did the authors employ a modularity cutoff for the Girvan-Newman method to control community subdivision? And if so, was the cutoff the same for each of the three mutant cases?

*Reviewer #2 (Recommendations for the authors):*

The paper would benefit from clarifying the importance of the structural dynamics to the specificity of the CRISP-Cas9 function. Specifically, the authors should explain whether changes in the intra- and inter-molecular communication are linked to a defined step of the nuclease (i.e., substrate recognition? chemical step? On and off rates?).

*Reviewer #3 (Recommendations for the authors):*

Figure 4B and 4C are never mentioned in the article, which I believe speaks to a bigger issue with the manuscript – a lot of NMR data has been collected, but it is not being utilized; its main role is to support the MD, but it is doing so only superficially. What are supposed to learn from the R1R2 product? There does not seem to be the expected increase in R1R2 values corresponding to residues that exhibit exchange. A line depicting the average or better the expected value of R1R2 would be informative, as the reader could then be able to pick out regions with slow and fast timescale dynamics. How do the order parameters or heteronuclear NOE compare with the MD? These timescales are definitely covered by the length of the simulation and may be more informative of the allosteric network than the CPMG data.

CPMG data were collected at multiple fields and data analyzed, yet nothing about these results are presented; what were the kex and pbs for these data? Do all the mutants populate the same excited state (as gauged from δ omegas)? How does this fit with the MD? I'm not convinced that "the allosteric signaling is preserved" in the mutants, and pointing to the shape and number of curves as evidence is not sufficient (i.e., lines 179-182). It's not clear that the dynamics measured by CPMG are on the same timescale as those measured by MD, even with a 3.6 us simulation. Fast pico-to-nanosecond timescale dynamics may be more informative (see above). Nevertheless, the mutants are clearly altering the micro-to-millisecond timescale dynamics (e.g., K782, E827, E873). Again, this is hardly discussed. Only a short acknowledgement is made about the larger R2,inf for K855A. What is the case for this?

The overall conclusion of the MD analysis seems at odds. The mutants maintain the same allosteric network but alter the network somehow to affect specificity. The network analysis shows a loss in communication between two allosteric communities, what does this mean exactly (line 223)? How does this loss disrupt allosteric cross-talk between RuvC and REC2 (line225)? The mutants increase communication between non-allosteric sites; is the loss or gain of communication observed in the NMR relaxation data?

---

## [Author Response]

Essential revisions:1) One significant concern raised by several reviewers is the lack of a direct comparison between the NMR studies and the MD simulations. This should be addressed in the revised manuscript.

This is indeed a very important point, which we are now addressing by analyzing the NMR relaxation data in the context of the communities. Specifically, we have evaluated how the dynamic exchange between the HNH communities is altered by the three K–to–A mutations, by employing an integrated approach introduced in a prior study of an allosteric enzyme (Lisi et al., PNAS 2017). In this analysis, the HNH communities identified through computational analysis are used as a reference, while the dynamic exchange among them is derived from CPMG relaxation dispersion (Figure 5). From this analysis of the NMR data, we observe a decrease in the dynamic exchange between the allosteric communities upon mutation, while an increase is observed in the non-allosteric communities. This reduction in the dynamic exchange observed in the mutants is consistent with the mutation-induced Edge Betweenness determined from computational analysis. We have now reported the outcomes of this analysis in the Results (pages 12-13, lines 250-256; 270-276) and in the Discussion (page 15, lines 316-318) sections, adding Figure 5 and Figure Supplement 9 and also reporting methodological details in the Methods section (page 24, lines 520-528).

2) The authors performed relaxation measurements for fast dynamics, however, they did not calculate the order parameters for the protein backbone. These should therefore be calculated and the authors should clearly indicate how the order parameters and heteronuclear NOEs compare to the calculated values from the MD trajectories.

The reviewer is absolutely correct and we have now included S^2^ parameters for each K-to-A mutant and determined the difference from WT HNH (new Figure Supplement 7). We also added discussion of these data on page 9, lines 187-192. Briefly, S^2^ parameters for each mutant are quite similar to those of WT HNH, evidenced by ΔS^2^ values ≤ 0.1 for the majority of residues. These data also mirror ΔS^2^ values determined from MD simulations. Further, we note agreement between S^2^ and the ^1^H-[^15^N] NOE that show depressed values sporadically between residues 800-825, surrounding residue 850, and at the C-terminus.

3) There is no mention of figures 4B and 4C in the manuscript. The results presented in these figures should be discussed, and protein regions showing slow and fast timescale dynamics should be clearly indicated.

We have corrected this oversight with new test explaining that K-to-A mutations do not introduce substantial changes to fast timescale dynamics in HNH (see page 9, lines 187-192). We note through additional analysis of T_1_, T_2_, 1H-[^15^N] NOE and order parameters (new Figure Supplement 7) that K-to-A mutants appear similar to WT HNH, mainly exhibiting fast dynamics sporadically between residues 800-825, surrounding residue 850, and at the C-terminus. We also note on page 9, line 173 several residues with CPMG relaxation dispersion, related to Figure 3A.

4) CPMG data were collected at multiple fields and the data analyzed, yet the analysis of these results was not presented. The authors should provide the kex and pbs obtained for these data, as well as indicate the changes between the wild type and mutant proteins. Do all the mutants populate the same excited state (as gauged from δ omegas)? How does this fit with the MD?

The Reviewer is correct that there is a much greater analysis that can be done. In addition to our new analysis of Rex and S2 parameters for all mutants (Figure Supplements 5, 7), we have updated/generated new Supplemental Tables S1-S3 to provide relaxation parameters for each of the mutants. We also analyzed the CPMG-derived kex values, noting that the mutants show relatively similar overall average kex, but skew the rates toward slower regimes when plotted as a distribution (new Figure Supplement 6 and text on page 9, lines 181-185). We also note the data are best described as individual, rather than global fits, as noted in J. Am. Chem. Soc. 2020, 142, 1348-1358 (page 20, line 420) and the average ground and excited state populations of the WT and HNH mutants are very similar (page 9, lines 185-187). Based on these data, and consistent with MD, it is not necessarily a large-scale global realignment of conformational exchange that affects specificity, but rather subtle changes in the location and nature of the dynamics that alters allosteric crosstalk.

5) The effect of the mutants on the micro-to-millisecond timescale dynamics should be discussed. Additionally, it is unclear how the dynamics or structural perturbations caused by these selected mutants are converted into the enzyme's increased or decreased specificity.

The Reviewer is correct that we have not adequately discussed this point. We include a new Figure 5 that connects the micro-millisecond motions detected by CPMG to changes in allosteric structure detected by simulations. Most critically, these new data show that dynamic exchange among allosteric communities (i.e., the number of CPMG active residues) is decreased in all of the K-to-A mutants. This is consistent with alterations of the dynamic structure that may account for changes in specificity.

We conducted further analysis of the R_ex_ parameter derived from CPMG experiments and note changes that may also inform the mechanism (new Figure Supplement 5). First, all K-to-A mutants display visually similar ΔR_ex_ profiles versus WT HNH, suggesting the mutations have a consistent effect on the protein despite occurring at different locations. This is consistent with small ΔR_ex_ when the mutants are compared to each other in the same figure. Second, the largest ΔR_ex_ occur in the regions of 780-790 and surrounding residue 825 (left panels of Figure Supplement 5). Our network analysis indicates these regions correspond to the A3 and A1 allosteric communities, respectively (Figures 4 and 6). Most notable is that ΔR_ex_ indicates a loss of flexibility in the mutants versus WT HNH (negative values plotted as mutant – WT). This is consistent with MD findings and is discussed on page 9, lines 175-179 and page 12, lines 233-237.

The reviewers also point out that the relation between alteration of the allosteric signaling and increase in specificity was not clearly discussed. We have now clarified this point at page 16 (lines 336-344) and briefly in the conclusions (page 19, lines 397-399), as summarized below. The HNH allosteric signaling connects the DNA recognition region (REC) and the catalytic sites (HNH and RuvC), transferring the information to DNA binding for cleavage. This REC to RuvC communication is a cornerstone of the CRISPR-Cas9 specificity. In fact, single-molecule experiments (Chen et al., Nature 2017, 550, 407–410; Dagdas et al., Sci. Adv. 2017, 3, eaao002) have shown that binding of off-target DNA sequences at the level of REC affects the dynamics of HNH and its allosteric activation of DNA cleavages. Our investigations show that the three enhanced specificity mutations disrupt this signaling mechanism that is critical for achieving specificity. Considering also that the magnitude of decrease in allosteric communication correlates to the order of specificity enhancement (K855A > K848A ~ K810A), the structural perturbations caused by the three K-to-A mutants can be related to the enzyme's increased specificity.

6) The authors state that the differences in the relaxation dispersion profiles are less than 1.5 Hz, indicating small changes in dynamics. A Plot showing the differences in the relaxation dispersion profiles (ΔRex) should be provided for all the proteins (WT and mutants) to support this claim.

We thank the Reviewer for pointing this out. We now include new Figure Supplement 5 that provides per-residue R_ex_ values, as well as ΔR_ex_. Our initial manuscript attempted to indicate that on average, changes in R_ex_ among the mutants were small (K855A-K848A <ΔR_ex_> = -0.65 s^-1^; K855A-K810A <ΔR_ex_> = -0.51 s^-1^; K848A-K810A <ΔR_ex_> = 0.21 s^-1^), which is now indicated on the right panels of Figure Supplement 5. On more careful analysis, we note some more significant changes that are quite interesting. First, all K-to-A mutants display visually similar ΔR_ex_ profiles versus WT HNH, suggesting the mutations have a consistent effect on the protein despite occurring at different locations. Second, the largest ΔR_ex_ occur in the regions of 780-790, 825, and 885 (left panels of Figure Supplement 5). Our network analysis indicates these regions correspond to A3, A1, and NA2, respectively. Most notable are the changes in A1 and A3, where ΔR_ex_ indicates a loss of flexibility in the mutants versus WT HNH, which is discussed on page 9.

7) The authors have chosen to use Grubmuller's generalized correlation to compute the weights on the nodes of the protein network. Grubmuller's generalized correlation captures both linear and non-linear correlations. Indeed, you could run the linearized version to distinguish the non-linear from linear correlations. Would the results have been different if Pearson correlation was used? Conversely, would there be key allosteric residues picked up by generalized correlation and not by linear correlation?

The choice of employing the Grubmuller's generalized correlation (GC) method was motivated by its capability to capture all types of correlations, which has shown in several reports to describe well the allosteric cross-talks in combination with network analysis (e.g., Rivalta et al., PNAS 2012, 109, 1428-1436; Melo et al., J. Chem. Phys. 2020, 153, 134104 and others).

To address the Reviewer’s comment, we have now computed the Pearson’s correlations (PCs) and compared them with the Grubmuller's GCs (Appendix figure 3). From our comparison, it is clear that the GC method can capture more correlations than the PCs. We also computed the allosteric pathways based on PCs (Appendix figure 4), which revealed a qualitative agreement with the pathways derived from GCs. This consistency shows that both methods convey in describing the systems’ correlations, yet the GC method is a more complete estimator of the overall coupled motions. Following this reasoning, we carried out the analysis of the allosteric mechanism based on the GCs.

8) When using the Girvan-Newman method to partition the network graph into communities, it is possible that the different simulation ensembles for the K855A, K810A and K484A mutants could result in different numbers of residues per community. In that case, it becomes difficult to compare the changes in betweenness for the mutants as there is also an accompanying shift in residues between communities. Could the authors please confirm that this is not the case and that each community contains the exact same number of residues for the K855A, K810A and K484A mutants.

The reviewer is correct. The number of residues per communities in the WT system and the K855A, K810A and K484A mutants slightly changes (Appendix figure 8). For this reason, to enable a proper comparison and, most importantly, to assess how the three K–to–A mutations alter the WT communication, the communities of the WT system have been used as a reference. This information was previously reported in the SI. We have now clarified this point in the main text (page 24, lines 518-521). For completeness, we have now also reported the community network plots and their corresponding Edge Betweennesses matrices in the SI (Appendix figure 8).

We also note that, in our revised manuscript, the communities of the WT HNH and of each mutant have been used as a reference to evaluate the dynamic exchange measured experimentally (Figure 5 and Figure Supplement 9). Further details are reported in the reply to point #1.

9) Did the authors employ a modularity cutoff for the Girvan-Newman method to control community subdivision? And if so, was the cutoff the same for each of the three mutant cases?

The *“modularity”* measures the strength (or the quality) of the community structure and is used to determine the optimal division of the network. In our application, the optimum community structures have the highest modularity of 0.7. This cutoff was applied for the WT system and for each of the three mutants, in line with our early study of the allosteric mechanism in Cas9 (Palermo et al., JACS 2017). This value is also consistent with modularity values found for other biomolecular systems of 0.4–0.8 (Saltalamacchia et al., JACS 2020, 142, 8403–8411). We have now discussed this point in the SI (*page S7*).

10) In the discussion, the authors refer to the synchronous motions that may be responsible for specificity. How did they deduce that the motions are synchronous? From MD simulations or the global fitting of the CPMG curves? Do motions need to be synchronous for effective allosteric communications?

In the manuscript, we referred to “synchronous” when describing community network analysis (CNA), where groups of residues displaying highly synchronized dynamics are gathered in communities. This wording is indeed employed in several computational studies harnessing CNA (*PNAS,* 2017, *114*, E3414-E3423). We therefore did not employ the word “synchronous” as mechanistically. We have now changed our phrasing in the manuscript to avoid any confusion.

11) The authors state that mutations can target sites identified in this study (hotspots) to improve CRISP-Cas9 function. Can the authors elaborate more on this point? How do they envision the mutations could tune the function of the complex?

We thank the reviewers for this insightful comment, which gives us the opportunity to suggest critical hotspots for mutational studies. Our computational analysis indicated that the three K–to–A mutations mainly disrupt the cross-talk between the A1 and A2 communities (Figure 4). This effect is observed for all mutants, and is confirmed by the analysis of the NMR relaxation data (Figure 5), suggesting that the A1-A2 communities are critical hotspots for the signal transmission. Building on this observation, mutational studies targeting residues of the A1-A2 communities could impact the allosteric communication and, in turn, modulate the function and specificity of the system. We have now included this discussion in the main text (pages 15-16, lines 318-324 and page 18, lines 375-377) adding Figure 6. The Abstract was also amended including this information.

12) Generally, there appear to be a number of grammatical and stylistic issues with the manuscript. Revisiting the writing could serve to improve the readability of the article.

We have gone through the manuscript and made changes to grammar and style where possible.